# Six-electron-conversion selenium cathodes stabilized by dead-selenium revitalizer for aqueous zinc batteries

Jingwei Du[1], Jiaxu Zhang[1], Xingyuan Chu[1], Hao Xu[1], Yirong Zhao[2], Markus Löffler[3], Gang Wang [1,4], Dongqi Li [1], Quanquan Guo[5], Ahiud Morag[1,5], Jie Du[1], Jianxin Zou[6], Daria Mikhailova[7], Vlastimil Mazánek[8], Zdeněk Sofer [8], Xinliang Feng [1,5] ✉ & Minghao Yu [1,5] ✉

Aqueous zinc batteries are attractive for large-scale energy storage due to their inherent safety and sustainability. However, their widespread application has been constrained by limited energy density, underscoring a high demand of advanced cathodes with large capacity and high redox potential. Here, we report a reversible high-capacity six-electron-conversion Se cathode undergoing a $ZnSe \leftrightarrow Se \leftrightarrow SeCl_4$ reaction, with $Br^-/Br_n^-$ redox couple effectively stabilizes the Zn||Se cell. This Se conversion, initiated in a $ZnCl_2$-based hydrogel electrolyte, presents rapid capacity decay (from 1937.3 to 394.1 mAh $g_{Se}^{-1}$ after only 50 cycles at 0.5 A $g_{Se}^{-1}$) primarily due to the dissolution of $SeCl_4$ and its subsequent migration to the Zn anode, resulting in dead Se passivation. To address this, we incorporate the $Br^-/Br_n^-$ redox couple into the Zn||Se cell by introducing bromide salt as an electrolyte additive. The generated $Br_n^-$ species acts as a dead-Se revitalizer by reacting with Se passivation on the Zn anode and regenerating active Se for the cathode reaction. Consequently, the cycling stability of the Zn||Se cell is improved, maintaining 1246.8 mAh $g_{Se}^{-1}$ after 50 cycles. Moreover, the Zn||Se cell exhibits a specific capacity of 2077.6 mAh $g_{Se}^{-1}$ and specific energy of 404.2 Wh $kg^{-1}$ based on the overall cell reaction.

Intensive research interest has been sparked by the feasibility of reversible electrochemical zinc (Zn) stripping/plating in mild acidic water-based electrolytes, driving the development of aqueous zinc batteries (AZBs)[1]. These batteries offer the promise of low cost, high safety, and desirable sustainability, making them particularly attractive for large-scale energy storage applications. Recent advancements have significantly mitigated issues associated with Zn metal anodes, such as dendrite formation, hydrogen evolution, surface corrosion, and passivation. These improvements have been achieved through a variety of strategies, including electrolyte optimization[2–4], interphase

[1]Faculty of Chemistry and Food Chemistry & Center for Advancing Electronics Dresden (cfaed), Technische Universität Dresden, Dresden 01062, Germany. [2]Physical Chemistry, Technische Universität Dresden, Zellescher Weg 19, Dresden 01069, Germany. [3]Dresden Center for Nanoanalysis (DCN), Center for Advancing Electronics Dresden (cfaed), Technische Universität Dresden, Helmholtzstraße 18, Dresden 01069, Germany. [4]Zhejiang Key Laboratory of Advanced Fuel Cells and Electrolyzers Technology, Materials Tech Laboratory for Hydrogen & Energy Storage, Ningbo Institute of Materials Technology and Engineering (NIMTE) of the Chinese Academy of Sciences, Ningbo 315201, China. [5]Max Planck Institute of Microstructure Physics, Weinberg 2, Halle 06120, Germany. [6]Center of Hydrogen Science & State Key Laboratory of Metal Matrix Composites, School of Materials Science and Engineering, Shanghai Jiao Tong University, Shanghai 200240, China. [7]Leibniz Institute for Solid State and Materials Research (IFW) Dresden e.V., Helmholtzstraße 20, Dresden 01069, Germany. [8]Department of Inorganic Chemistry, Faculty of Chemical Technology, University of Chemistry and Technology Prague, Technická 5, Prague 6 16628, Czech Republic. ✉e-mail: xinliang.feng@tu-dresden.de; minghao.yu@tu-dresden.de

construction[5–9], and electrode morphology/structure engineering[10–12]. Conversely, the energy density, a key factor determining the market scope of batteries, largely depends on the choice of cathode (positive electrodes) materials when using Zn metal as the anode (negative electrodes). Intercalation-type manganese (Mn)-based and vanadium (V)-based oxides emerge as promising high-energy cathode options, offering specific capacities ranging from 200–600 mAh g$^{-1}$ and a decent midpoint discharge voltage of 0.7~1.3 V[13–15]. These metrics translate to an specific energy of up to 210 Wh kg$^{-1}$ based on the overall cell reaction. Further advancements in the energy density of AZBs are highly desired and necessitate the discovery of novel cathode chemistries with larger specific capacities and higher operating potentials.

Unlike intercalation-type electrodes, conversion-type electrodes undergo distinct phase transformations during charge and discharge, enabling multiple electron transfer and relatively high specific capacities[16,17]. As a representative example, sulfur (S) has been regarded as a promising high-energy cathode option for AZBs based on a two-electron S/ZnS conversion process, offering a theoretical specific capacity of 1675 mAh g$^{-1}$[18,19]. However, the practicality of Zn||S batteries is hindered by severe kinetics issues caused by the insulating nature of S, along with the relatively low voltage (0.63 V), limiting the achieved specific energy to below 730 Wh kg$_S^{-1}$[18]. Selenium (Se), another chalcogen element, exhibits significantly higher intrinsic conductivity than S ($1 \times 10^{-3}$ S m$^{-1}$ vs. $5 \times 10^{-28}$ S m$^{-1}$) and has recently been proposed for a six-electron Se$^{2-}$/Se$^{0}$/Se$^{4+}$ conversion reaction in battery cathodes[20,21]. With such a conversion mechanism, Se presents a theoretical capacity of 2038 mAh g$^{-1}$ and 9759 mAh cm$^{-3}$, which are about 1.2-fold and triple those of two-electron-conversion S electrodes (1675 mAh g$^{-1}$ and 3467 mAh cm$^{-3}$), respectively. Coupled with the high redox potential of Se$^{0}$/Se$^{4+}$ (0.74 vs. standard hydrogen electrode, ca. 1.5 V vs. Zn/Zn$^{2+}$), Zn||Se batteries are expected to considerably surpass Zn||S batteries in terms of energy density. The first demonstrated aqueous Zn||Se batteries employed a Zn di[bis(trifluoromethylsulfonyl)imide]-based electrolyte, delivering a specific capacity of 611 mAh g$_{Se}^{-1}$ and an average discharge voltage of 1.23 V[22]. While the prevailing conversion reaction entailed the two-electron Se/ZnSe conversion, minimal Se$^{0}$/Se$^{4+}$ conversion was detected[23]. Moreover, a six-electron Se conversion undergoing an overall ZnSe↔Se↔SeO$_2$ reaction was identified in Zn trifluoromethanesulfonate (Zn(OTf)$_2$)-based aqueous electrolytes with Cu[Co(CN)$_6$] as the catalytic host, yielding a specific capacity of 664.7 mAh g$_{Se}^{-1}$ and an average discharge voltage of 1.1 V[21]. Additionally, an amphoteric redox reaction of Se was also confirmed for organic triphenylphosphine selenide in Zn(OTf)$_2$-based electrolyte[24]. Nevertheless, the conversion depth of Se achieved so far remains significantly limited (<15.5%) compared to the theoretical promise in specific capacity. This obstacle could be attributed to the retarded Se$^{0}$/Se$^{4+}$ conversion kinetics and efficiency, potentially induced by the involvement of bulky anions.

In this study, we achieve the six-electron conversion of Se for high-energy AZBs through a ZnSe↔Se↔SeCl$_4$ process as Eqs. (1–3), wherein the Br$^-$/Br$_n^-$ ($3 \leq n \leq 7$) redox couple plays an essential role as the inactive "dead" Se revitalizer to stabilize the electrode performance. The Se conversion is firstly initiated in a ZnCl$_2$-based hydrogel electrolyte (denoted ZCE), exhibiting rapid capacity decay from 1937.3 to 394.1 mAh g$_{Se}^{-1}$ after only 50 cycles. We identify that this decay primarily originates from the dissolution of the charging product (i.e., SeCl$_4$) into the electrolyte, leading to its subsequent migration to the Zn anode and causing dead Se passivation (Fig. 1a). By introducing bromide salt additive (e.g., tetraethylammonium bromide (Et$_4$NBr)) into ZCE (denoted ZCE-Br), the cycling stability of the Zn||Se cell is considerably improved with 1246.8 mAh g$_{Se}^{-1}$ maintained after 50 cycles. We disclose that the Br$^-$/Br$_n^-$ redox couple is incorporated into the Zn||Se cell, with the generated Br$_n^-$ species acting as the dead-Se revitalizer by reacting with Se passivation on the Zn anode and regenerating active Se for the cathode reaction (Fig. 1b). Additionally, the Zn||Se cell delivers a maximum specific capacity of 2077.6 mAh g$_{Se}^{-1}$ and a specific energy of 404.2 Wh kg$^{-1}$ based on the overall cell reaction, including the mass of active anode/cathode materials and the consumed electrolyte salt.

Se cathode reactions:

$$Se + 4Cl^- \rightleftharpoons SeCl_4 + 4e^- \tag{1}$$

$$Se + Zn^{2+} + 2e^- \rightleftharpoons ZnSe \tag{2}$$

Overall cell reaction:

$$ZnSe + 2ZnCl_2 \rightleftharpoons SeCl_4 + 3Zn \tag{3}$$

## Results

### Six-electron Se conversion

Two characteristics of aqueous electrolytes are considered essential for reversible six-electron Se conversion: (1) small anion charge carriers to initiate high-kinetics Se$^{0}$/Se$^{4+}$ conversion, and (2) low water reactivity to support high operating voltage and inhibit Se$^{4+}$ hydrolysis. In this regard, we first utilized the ZCE electrolyte composed of 30 m ZnCl$_2$ and poly(ethylene oxide) as the base electrolyte to evaluate the electrochemical performance of the Se cathode in a two-electrode Swagelok cell with Zn foil as the counter electrode. The Se cathode was fabricated by melting Se into activated carbon (AC, Supplementary Figs. 1–2). Figure 2a displays the initial three cyclic voltammetry (CV)

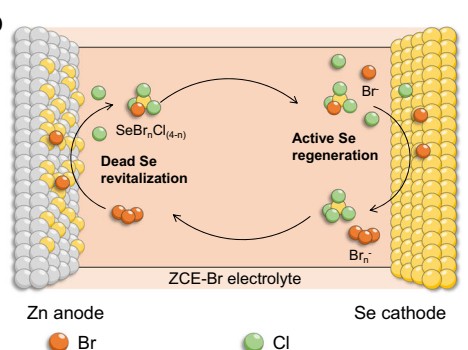

**Fig. 1 | Schematic showing the configuration of the Zn||Se cells. a** Zn||Se cell with ZCE (30 m ZnCl$_2$ hydrogel electrolyte), in which rapid performance decay is caused by the formation of dead Se on Zn anode. **b** Zn||Se cell with ZCE-Br (30 m ZnCl$_2$ hydrogel electrolyte with bromide salt additive), in which the Br$^-$/Br$_n^-$ redox couple serves as a dead Se revitalizer and stabilizes the Se electrode performance.

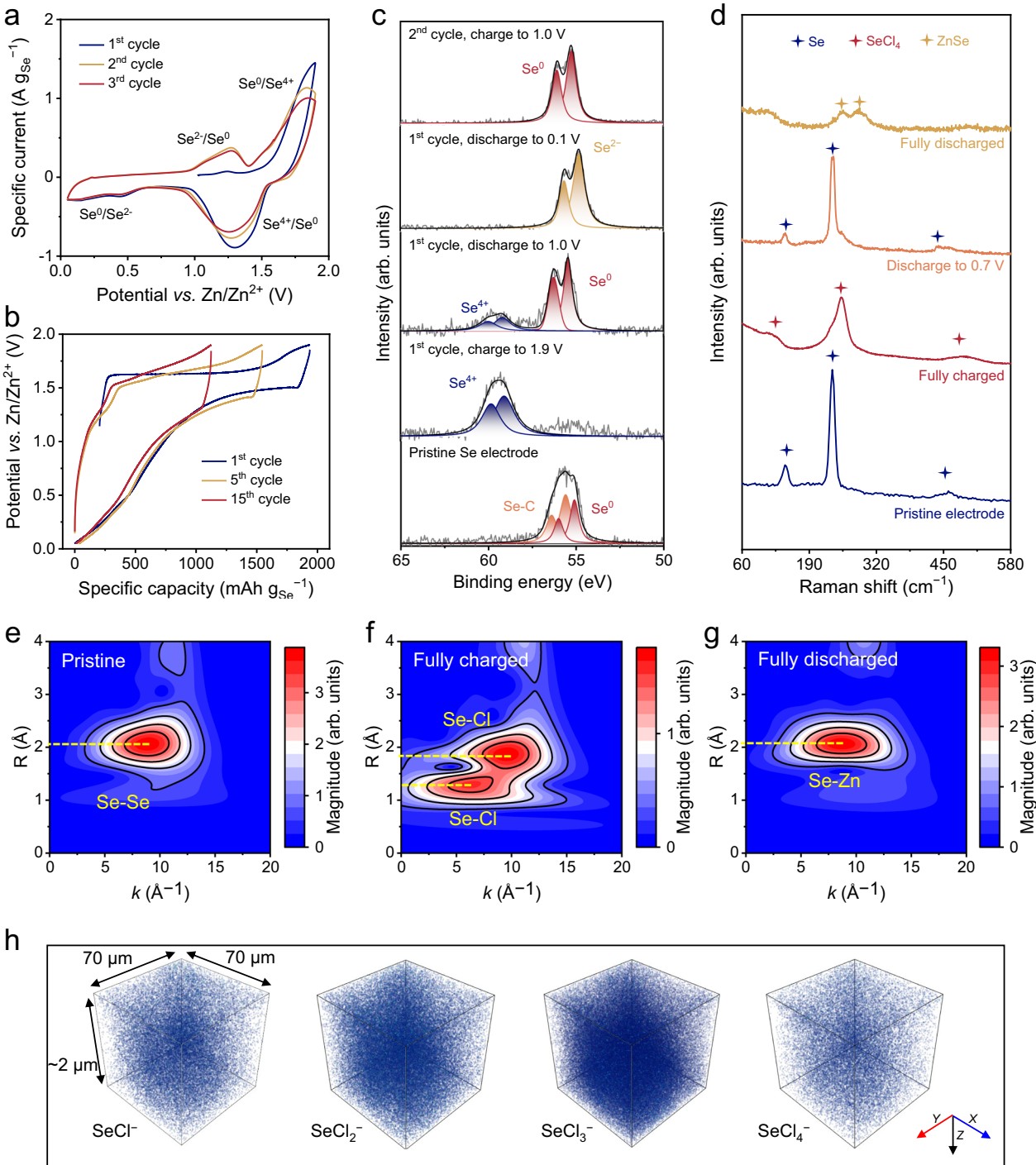

**Fig. 2 | Mechanism elucidation of six-electron Se conversion. a** The initial three-cycle CV curves of the Se electrode at 0.1 mV s$^{-1}$. **b** The first-, fifth- and fifteenth-cycle GCD profiles of the Se electrode at 0.5 A g$_{Se}$$^{-1}$. **c** Se 3$d$ XPS spectra and (**d**) Raman spectra of the Se electrode at different charge/discharge states. Wavelet-transformed Se K-edge EXAFS of the Se electrode at (**e**) the pristine state, (**f**) fully charged state, and (**g**) fully discharged state. **h** 3D reconstructions of the TOF-SIMS signals for SeCl$_4$$^-$, SeCl$_3$$^-$, SeCl$_2$$^-$, and SeCl$^-$ in the Se electrode at the fully charged state.

cycles of the Se cathode, revealing an open circuit potential of 1.02 V *vs*. Zn/Zn$^{2+}$. During the first anodic scan, the electrode showed a sharp oxidation peak, indicating the initiation of Se$^0$/Se$^{4+}$ conversion. In the subsequent cathodic scan, two reduction zones were observed at 0.75 ~ 1.55 V and 0.05 ~ 0.75 V, likely corresponding to Se$^{4+}$/Se$^0$ and Se$^0$/Se$^{2-}$ reduction, respectively. In the following 2nd and 3rd cycle, the Se electrode maintained these two redox zones, implying reversible conversion of Se$^{2-}$/Se$^0$/Se$^{4+}$.

Galvanostatic charge-discharge (GCD) profiles at 0.5 A g$_{Se}$$^{-1}$ were further collected to quantify the specific capacity of the Se electrode (Fig. 2b and Supplementary Fig. 3). After the first-cycle activation, the Se electrode achieved a high specific capacity of 1937.3 mAh g$_{Se}$$^{-1}$. By subtracting the capacity contributed from AC (Supplementary Fig. 4), a specific capacity of 1767.3 mAh g$_{Se}$$^{-1}$ was estimated for the Se conversion reaction, referring to ~5.2-electron transfer per Se atom. The discharge curve features a plateau at around 1.5 V and a

prolonged slope, consistent with the CV shape. However, the capacity of the Se electrode decayed rapidly, with only 1510.5 mAh $g_{se}^{-1}$ and 1113.1 mAh $g_{Se}^{-1}$ retained at the 5th and 15th GCD cycle.

We then investigated the Se conversion mechanism by characterizing the Se electrode at different charge/discharge states. Se 3d X-ray photoelectron spectroscopy (XPS) spectra (Fig. 2c) revealed that the electrode was converted from elemental $Se^0$ with characteristic peaks at 56.0 eV (Se $3d_{3/2}$) and 55.1 eV (Se $3d_{5/2}$) to $Se^{4+}$ with characteristic peaks at 59.9 eV (Se $3d_{3/2}$) and 59.1 eV (Se $3d_{5/2}$) during the initial charge to 1.9 V[21,25]. In comparison with the $Se^0$ peaks, the slightly higher binding energy of the Se-C peaks in the pristine Se electrode can be attributed to the melt-diffusion process of Se into AC (at 600 °C), which triggered the formation of interfacial Se-C bonds[26]. In subsequent cycles, the conditions required for the reformation of the Se-C bond, such as the high-temperature melt-diffusion process, are absent. Therefore, the disappearance of the Se-C bond has no impact on the electrochemical performance of the system. Upon discharging to 1.0 V, the $Se^0$ peaks became considerably intensified, while the $Se^{4+}$ peak diminished, indicating a reverse to $Se^0/Se^{4+}$ conversion. Further discharging the electrode to 0.1 V shifted the Se $3d_{3/2}$ and $3d_{5/2}$ peaks to lower binding energy of 55.6 eV and 54.7 eV, respectively, aligning with characteristic $Se^{2-}$ peaks[21]. Besides, $Se^{2-}$ can be re-oxidized to $Se^0$ by charging the electrode to 1.0 V. These XPS findings signify the reversible six-electron $Se^{2-}/Se^0/Se^{4+}$ conversion of the Se electrode.

Figure 2d compares the Raman spectra of the Se electrode at varying charge and discharge states. Importantly, the fully charged electrode exhibits pronounced peaks at 126 $cm^{-1}$, 253 $cm^{-1}$, and a broaden peak ranging from 474 $cm^{-1}$ to 515 $cm^{-1}$, corresponding to characteristic peaks of $SeCl_4$[27,28]. Distinctively, the fully discharged electrode presents the characteristic peaks of ZnSe at 251 $cm^{-1}$ and 290 $cm^{-1}$[29]. At an intermediate charge state (i.e., 0.7 V vs. Zn/Zn$^{2+}$), characteristic Se peaks at 145 $cm^{-1}$, 236 $cm^{-1}$, and 460 $cm^{-1}$[30] are underscored. These Raman results further validate the $Se^{2-}/Se^0/Se^{4+}$ conversion and confirm the detailed conversion reaction as $ZnSe \leftrightarrow Se \leftrightarrow SeCl_4$ through Eqs. (1–2). Of note, a similar $ZnSe \leftrightarrow Se \leftrightarrow SeCl_4$ conversion was recently identified in a non-aqueous ionic liquid electrolyte[31]. However, the reported electrode exhibited low Se utilization efficiency (<700 mAh $g^{-1}$), in stark contrast to the high efficiency achieved here in an aqueous electrolyte.

Additionally, Se K-edge X-ray absorption spectroscopy provides essential insights into the local coordination environment of Se atoms in the fully charged and discharged electrodes (Supplementary Fig. 5). The wavelet transform (WT) analysis on the $k^3$-weighted EXAFS data with a 2D representation in R and k spaces was conducted to qualitatively evaluate the backscattering atoms of Se (Fig. 2e-g). For pristine Se (Fig. 2e), the WT maxima position is located at R = 2.05 Å, suggesting Se-Se scattering (Supplementary Fig. 6). At the fully charged state (Fig. 2f), new signals at R = 1.27 Å and R = 1.85 Å appear, indicating Se-Cl scattering, consistent with the maxima positions of $SeCl_4$ (Supplementary Fig. 6). This finding is further verified through time-of-flight secondary ion mass spectrometry (TOF-SIMS, Supplementary Fig. 7). A 3D reconstruction of TOF-SIMS signals from the Se cathode at the fully charged state (Fig. 2h) clearly illustrates a homogeneous distribution of Se-Cl species in the electrode, implying $Cl^-$ anions as charge carriers. At the fully discharged state (Fig. 2g), the WT maxima is at around R = 2.10 Å, consistent with Se-Zn scattering of ZnSe (Supplementary Fig. 6e), indicating ZnSe as the fully discharged product.

### Stabilizing the ZnSe ↔ Se ↔ SeCl₄ conversion

To understand the rapid capacity degradation observed for the Se electrode, we assembled the Zn||Se cell with the 30 m $ZnCl_2$ aqueous electrolyte (denoted a-ZCE). Instead of ZCE, a-ZCE was used to simplify the characterization of the Zn metal surface by preventing contamination from poly(ethylene oxide) presented in ZCE. After 20 charge/discharge cycles at 0.5 A $g_{se}^{-1}$, the cell was disassembled for

further analysis. A striking color change was observed for the Zn anode from its original silver color to red, indicating drastic surface passivation (Fig. 3a). SEM image of the Zn anode revealed a rough surface with inhomogeneous sediment, and corresponding energy dispersive X-ray spectroscopy (EDX) elemental mapping images confirmed the presence of Se-based species on the Zn anode (Fig. 3b). This nonnegligible Se signal was further corroborated by the derived EDX spectrum (Fig. 3c).

Se 3d XPS spectrum (Fig. 3d) was collected to determine the constituents of the red deposit sediment, unveiling that Se-based species on the Zn anode predominantly consist of elemental Se, with a minor presence of $Se^{4+}$ compound. This finding was further supported by Raman spectra (Fig. 3e), where three spots were selected on the cycled Zn anode to thoroughly assess the components. Specifically, both Se (characteristic peaks at 145 and 236 $cm^{-1}$) and $SeCl_4$ (characteristic peaks at 126, 253, and 474 $cm^{-1}$) were detected. Based on this observation, we infer that the primary cause of Zn anode passivation originates from the shuttling of the charged product, $SeCl_4$. Consequently, $SeCl_4$ underwent reduction on the anode surface, leading to the formation of inactive 'dead' Se through Eq. (4), accelerating the irreversible loss of active materials in the cathode. Moreover, the high solubility of commercial $SeCl_4$ in a-ZCE was also identified (Supplementary Fig. 8). Owing to the reduced water reactivity in a-ZCE, no hydrolysis of $SeCl_4$ was detected (Supplementary Fig. 9). To validate our inference, we assembled a Zn||AC cell using 30 m $ZnCl_2$ + 0.1 m $SeCl_4$ as the electrolyte. After 20 CV cycles at 1 mV $s^{-1}$, dead Se presented on the Zn anode (Supplementary Fig. 10).

$$SeCl_4 + 4e^- \rightarrow Se + 4Cl^- \qquad (4)$$

Preventing the formation of dead Se is crucial for enhancing the cycling stability of the Se electrode. In this context, the incorporation of strongly oxidative species capable of oxidizing and removing dead Se could be highly beneficial. We found that introducing the $Br^-/Br_n^-$ redox couple to the Zn||Se cell through the addition of bromide salts (e.g., $Et_4NBr$, $ZnBr_2$, or LiBr) as the electrolyte additive was an effective approach in stabilizing the Se conversion reaction (Supplementary Fig. 11). All three types of bromide salts demonstrated efficacy in enhancing the cycling stability of the $ZnSe \leftrightarrow Se \leftrightarrow SeCl_4$ conversion. The optimal electrolyte, 0.1 m $Et_4NBr$-added ZCE (i.e., ZCE-Br), enabled the Zn||Se cell to retain 61.5% of its capacity after 50 charge/discharge cycles at 0.5 A $g_{se}^{-1}$, in stark contrast to the ZCE electrolyte, which retained only 20.3%. Compared with ZCE, ZCE-Br resulted in only a slightly increased self-discharge issue for the Zn||Se cell (Supplementary Fig. 12). We also optimized the concentration of $Et_4NBr$ and observed that a higher $Et_4NBr$ concentration led to a lower coulombic efficiency in the Zn||Se cell, as more $Br^-/Br_n^-$ redox reactions were involved (Supplementary Fig. 13).

We evaluated the Zn anodes in the Zn||Se cells using the ZCE and ZCE-Br electrolytes after 10 charge/discharge cycles by SEM and elemental analysis. Unlike the cycled anode in ZCE, which exhibits an uneven and corrosive surface morphology (Fig. 3f), the cycled anode in ZCE-Br shows a notably smoother surface (Fig. 3g). Furthermore, in contrast to a distinct Se signal detected from the cycled anode in ZCE (Fig. 3h), there is barely a visible Se signal from the cycled anode in ZCE-Br, which clarifies the successful inhibition of dead Se formation during charging and discharging (Fig. 3i). A similar conclusion was drawn from the Raman and XPS analyses of the cycled Zn anodes in both ZCE and ZCE-Br (Supplementary Fig. 14).

### The Br⁻/Brₙ⁻ redox couple as the dead-Se revitalizer

To understand the impact of the bromide additive, we initially compared the CV profiles of the Se electrode in ZCE and ZCE-Br at 0.1 mV $s^{-1}$ (Fig. 4a and Supplementary Fig. 15). Aside from an additional reduction peak at 1.67 V observed in ZCE-Br, the two CV profiles

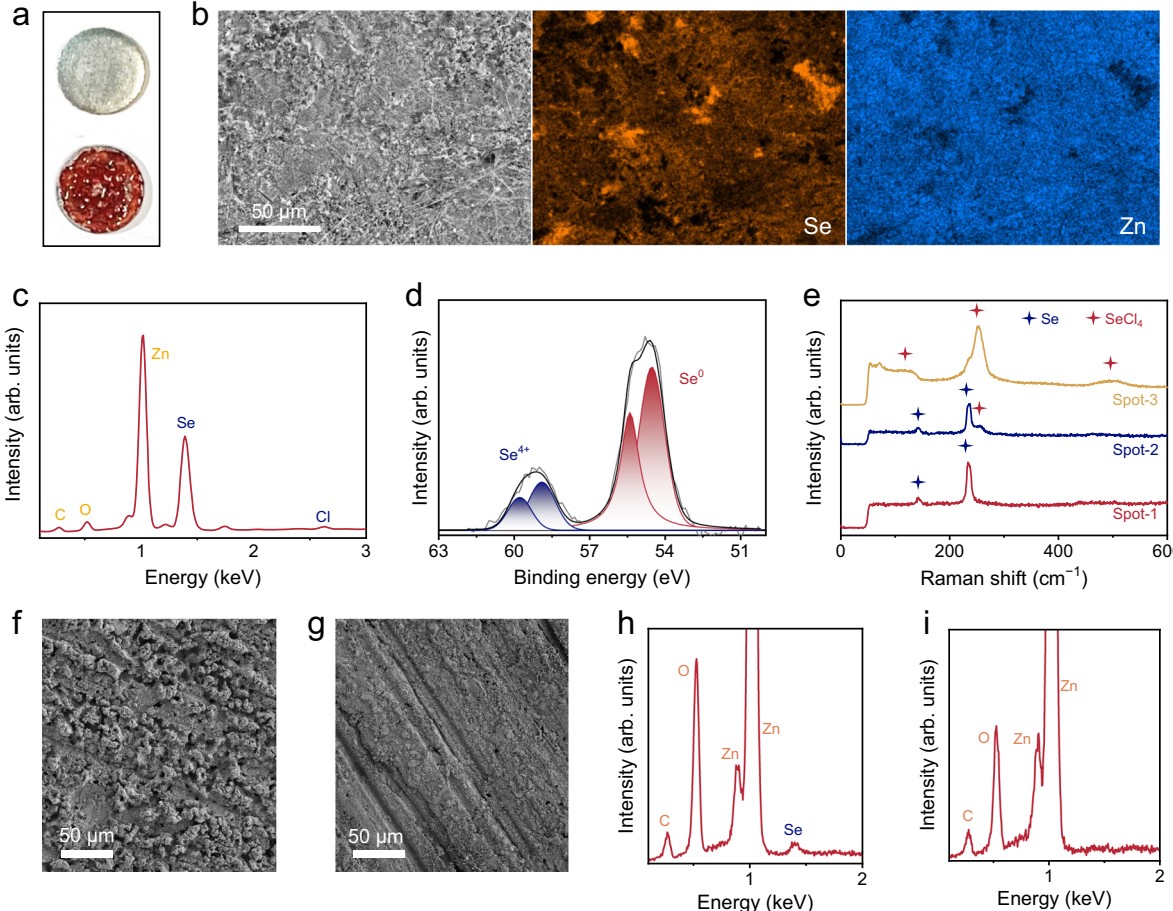

**Fig. 3 | Characterizations of the Zn anode in Zn||Se cell. a** Photos of the Zn anode in Zn||Se cell with a-ZCE before (up) and after (down) 20 cycles at 0.5 A g$_{Se}^{-1}$. **b** SEM image and corresponding EDX elemental mapping images, (**c**) EDX, (**d**) Se 3$d$ XPS, and (**e**) Raman spectra of the Zn anode after cycling in Zn||Se cell with a-ZCE. SEM images of the Zn anodes in Zn||Se cells with (**f**) ZCE and (**g**) ZCE-Br after 10 cycles at 0.5 A g$_{Se}^{-1}$. EDX spectra of the Zn anodes in Zn||Se cells with (**h**) ZCE and (**i**) ZCE-Br after 10 cycles at 0.5 A g$_{Se}^{-1}$.

display similar shapes, suggesting that the overall charge storage of the Se electrode was primarily governed by the ZnSe↔Se↔SeCl$_4$ conversion in both electrolytes. The minor reduction peak at 1.67 V is indicative of the presence of the Br$^-$/Br$_n^-$ couple, as depicted in Eq. (5)[32]. The high reversibility of the Br$^-$/Br$_n^-$ couple was evidenced by the performance of the Zn||AC cell with ZCE-Br (Supplementary Fig. 16). Moreover, we verified the comparable ionic conductivity (Supplementary Fig. 17) and Zn stripping/platting performance (Supplementary Fig. 18) between ZCE and ZCE-Br.

$$Br_n^- + (n-1)e^- \rightleftharpoons nBr^- \ (3 \le n \le 7) \tag{5}$$

Br K-edge XANES spectra of the Zn||Se cell with 0.1 m Et$_4$NBr-added a-ZCE (denoted a-ZCE-Br) further corroborate the existence of the Br$^-$/Br$_n^-$ couple (Fig. 4b). At the fully charged state, the Br K-edge XANES spectrum of the cell showed a negatively shifted adsorption edge compared to the spectrum of the as-fabricated cell, evidencing the oxidation of Br$^-$ anions[33]. Moreover, a pronounced peak at 13473 eV was observed for the charged cell, corresponding to the electron transition from 1$s$ to 4$p$. The emergence of this peak signifies the generation of holes in Br 4p orbitals, indicative of Br$_n^-$ species. Additionally, Se K-edge X-ray adsorption spectra were collected and compared for the fully charged Se electrodes in a-ZCE and a-ZCE-Br (Supplementary Fig. 19). The wavelet transform analysis on the $k^3$-weighted EXAFS data with a 2D representation in $R$ and $k$ spaces was

conducted for both electrodes. For the Se electrode charged in a-ZCE (Fig. 2f), the WT maxima at $R = 1.27$ Å represents the first-shell Se-Cl scattering, showing a $k$-value of 6.2 Å$^{-1}$. In comparison, the first-shell scattering extends towards the high-$k$ direction (10.3 Å$^{-1}$) for the Se electrode charged in a-ZCE-Br (Fig. 4c), implying the bonding between Se with heavier atoms than Cl[34]. This finding suggests the slight participation of Br$^-$ or Br$_n^-$ as charge carriers for the Se/Se$^{4+}$ conversion. It is worth noting that SeCl$_4$ remains the predominant charge product for the Se electrode in a-ZCE-Br, as supported by the Raman (Supplementary Fig. 20) and XPS (Supplementary Fig. 21) analyses.

Typically, strongly oxidative Br$_n^-$ species are soluble in aqueous solutions and can directly react with elemental Se in ambient environment through Eq. (6)[35,36]. We utilized commercial Se and Br$_2$ aqueous solution to verify their chemical reaction with a soluble product (Supplementary Fig. 22). Specifically, upon adding excess Br$_2$ aqueous solution to a vial containing Se, the solution became shallow in color, along with the complete consumption of Se. Conversely, when Br$_2$ aqueous solution was introduced to a vial containing excess Se, the solution turned colorless and transparent, indicating the complete consumption of Br$_2$. Given the abundant polybromide species in Br$_2$ aqueous solution[37,38], we deduced that Br$_n^-$ could act as an effective dead-Se revitalizer, reacting with the Se passivation layer on the Zn surface and re-activating Se for the cathode reaction.

$$4 Br_n^- + (n-1)Se \rightarrow (n-1)SeBr_4 + 4 Br^- \ (3 \le n \le 7) \tag{6}$$

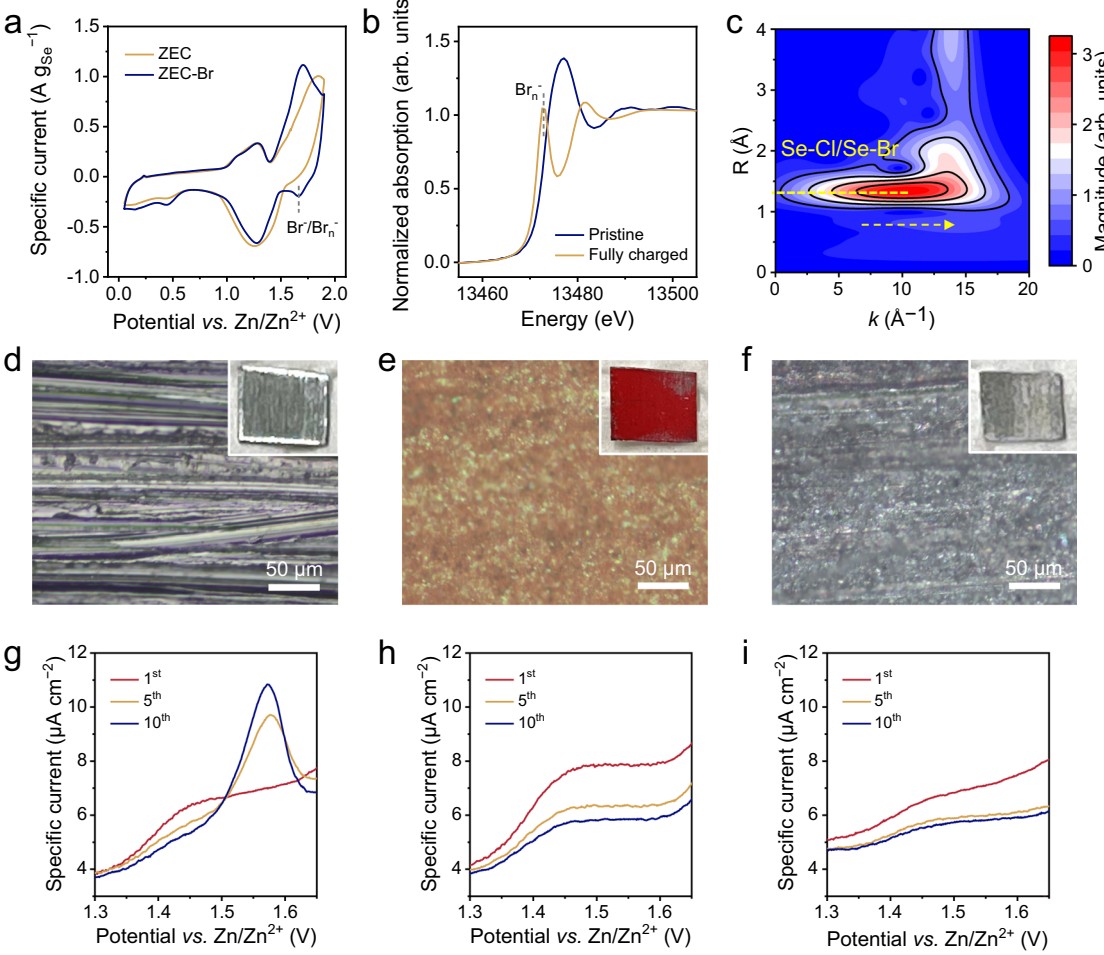

**Fig. 4 | Mechanism elucidation of the Br$^-$/Br$_n^-$ redox couple. a** CV profiles of the Zn||Se cells with ZCE and ZCE-Br at 0.1 mV s$^{-1}$. **b** Br K-edge XANES spectra of the Zn||Se cell with a-ZCE-Br at the pristine and fully charged states. The XANES spectra were normalized with pre-edge and post-edge. **c** Wavelet-transformed Se K-edge EXAFS of the fully charged Se electrode in a-ZCE-Br. Optical microscope images of (**d**) the fresh Zn anode, (**e**) Se@Zn anode, and (**f**) Se@Zn anode after cycling in a-ZCE-Br. The insets show the corresponding digital photos. Anodic part of the CV curves at 1 mV s$^{-1}$ of the (**g**) Se@Zn||Ti cell with a-ZCE-Br, (**h**) the Zn||Ti cell with a-ZCE-Br, and (**i**) the Se@Zn||Ti cell with a-ZCE.

To prove our hypothesis, Raman spectra were collected for pristine a-ZCE-Br and a-ZCE-Br in the Zn||Se cell after the first charge cycle (Supplementary Fig. 23). The dissolution of Br$_n^-$ in a-ZCE-Br was verified by the characteristic Raman peaks at 237 cm$^{-1}$ and 256 cm$^{-1}$[139]. Besides, optical microscope images of the Zn anode under different conditions were captured. After polishing with abrasive paper, fresh Zn foil exhibited a metallic luster with straight grinding trace (Fig. 4d). When this fresh Zn foil was assembled in a Zn||Se cell with a-ZCE and subjected to ten GCD cycles at 0.5 A g$_{Se}^{-1}$, the metallic luster of Zn foil and grinding traces disappeared, leading to a Se-passivated Zn anode (denoted Se@Zn), uncovering a severe shuttle effect during charging and discharging (Fig. 4e). To evaluate the effect of Br-modified electrolyte, Se@Zn was further employed as the anode in a-ZCE-Br, with AC as cathode. After fifty cycles at 0.5 A g$_{AC}^{-1}$, the optical microscope image of the Zn anode re-exhibited shallow grinding traces without distinct red precipitate (Fig. 4f), suggesting that the dead Se was wiped out during charging and discharging in a-ZCE-Br.

We next assembled a Se@Zn||Ti cell utilizing Ti mesh as the working electrode and a-ZCE-Br as the electrolyte, in order to corroborate the revitalization of dead Se. For comparison, control cells were also assembled, including Zn||Ti with the a-ZCE-Br electrolyte and Se@Zn||Ti cell with the a-ZCE electrolyte. CV curves of the three cells at 1 mV s$^{-1}$ were compared (Supplementary Fig. 24). Figure 4g zooms into the anodic scan of the Se@Zn||Ti cell with a-ZCE-Br in the

potential range of 1.30 ~ 1.65 V vs. Zn/Zn$^{2+}$. In the first CV cycle, the cell presented only one oxidation peak at 1.76 V, which can be attributed to the combination of Br$^-$/Br$_n^-$ and slight oxidation of the Ti mesh substrate. After five activation cycles, another distinct oxidation peak emerged at 1.58 V, consistent with the Se/Se$^{4+}$ conversion. This peak further intensified after ten CV cycles, underscoring the revitalization of dead Se on the Se@Zn anode. Such a revitalization behavior was not observed for the Zn||Ti cell with a-ZCE-Br (Fig. 4h) and the Se@Zn||Ti cell with a-ZCE (Fig. 4i). Besides, to evaluate effectiveness of this strategy for long-term cycling, the Zn anode in Zn||Se cell with the ZCE-Br electrolyte was investigated with elemental analysis after 200 cycles under specific current of 0.5 A g$_{Se}^{-1}$ (Supplementary Fig. 25). No obvious Se signal was detected, suggesting that the absence of dead Se on the Zn anode.

Above investigations conclude that the Br$^-$/Br$_n^-$ redox couple, acting as the dead-Se revitalizer, effectively avoids the passivation of Zn anode by the shuttling of SeCl$_4$ and the subsequent loss of active cathode material. This effect results in a considerably stabilized ZnSe↔Se↔SeCl$_4$ conversion in the ZCE-Br electrolyte.

## Performance of the stabilized Zn||Se cell

With the ZCE-Br electrolyte stabilizing the ZnSe↔Se↔SeCl$_4$ conversion, we systematically evaluated the electrochemical performance of the Zn||Se cell. Figure 5a and Supplementary Fig. 26 displays the GCD

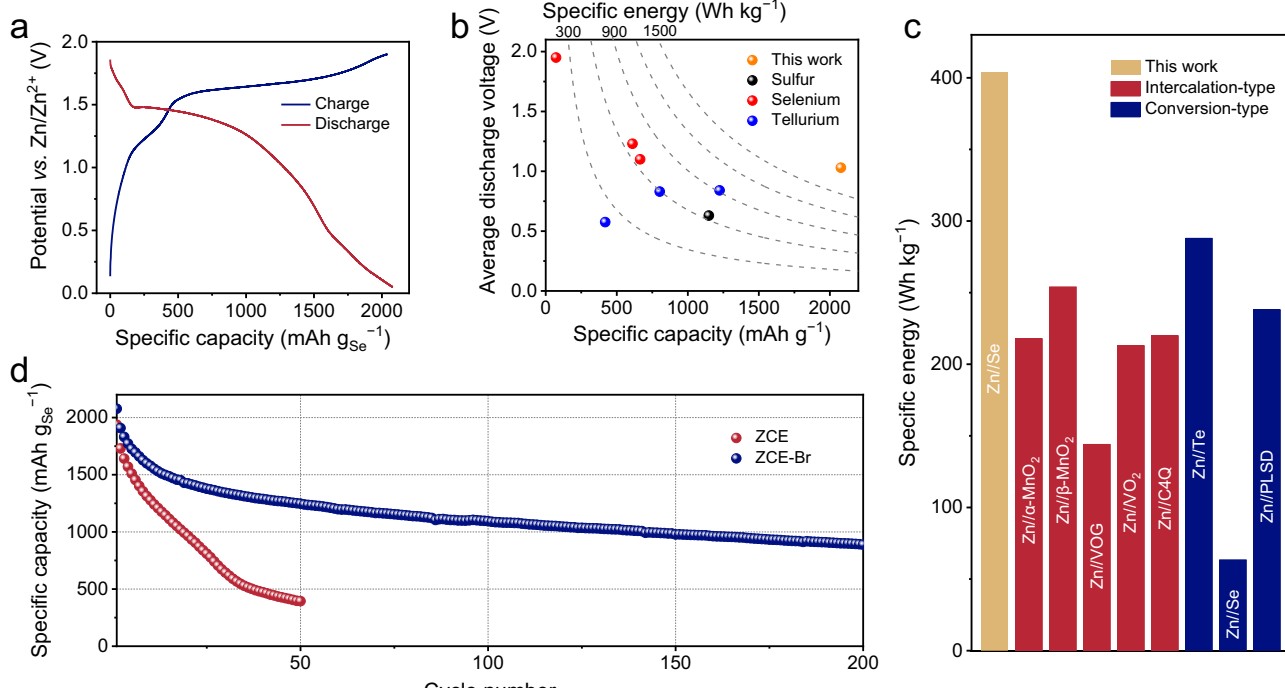

**Fig. 5 | Electrochemical performance of the Zn||Se cell with ZCE-Br. a** GCD profile at 0.5 A $g_{Se}^{-1}$. **b** Comparison of performance (encompassing gravimetric capacity, average discharge voltage, and gravimetric specific energy) of six-electron Se conversion chemistry with reported chalcogen cathodes of AZBs. **c** Comparison of specific energy between the Zn||Se device and recently reported AZB devices. The specific energy was calculated considering the entire battery reaction, including active anode/cathode materials and the consumed electrolyte salt. The source of the literature data shown in this figure can be found in Supplementary Information, Table 2. **d** Cycling performance of Zn||Se cells with different electrolytes at 0.5 A $g_{Se}^{-1}$.

curve of the Zn||Se cell at 0.5 A $g_{Se}^{-1}$, which points to a discharge capacity of 2077.6 mAh $g_{Se}^{-1}$. By subtracting the capacity contribution from the $Br^-/Br_n^-$ couple and the capacitive charge storage of AC (Supplementary Fig. 27), a specific capacity of 1731.5 mAh $g_{Se}^{-1}$ was estimated for the Se conversion reaction, well comparable to the Se electrode in the ZCE electrolyte. This charge-storage capacity translates to a maximum specific energy of 2138.5 Wh $kg_{Se}^{-1}$ (Fig. 5b), representing a competitive value among all reported conversion-type chalcogen cathodes, including S (260~725 Wh $kg_S^{-1}$)[18], Se (621~752 Wh $kg_{Se}^{-1}$)[21,22,24], and Te (234~1028 Wh $kg_{Te}^{-1}$)[40-42], as well as other reported AZB cathodes (Supplementary Table 1). Additionally, GCD profiles of our Zn||Se cell were collected with a potential window of 0.6~1.9 V vs. $Zn/Zn^{2+}$, utilizing only $Se^0/Se^{4+}$ conversion for charge storage (Supplementary Fig. 28). The corresponding specific capacity and specific energy were calculated as 1361.7 mA $g_{Se}^{-1}$ and 1801.6 Wh $kg_{Se}^{-1}$, accounting for 73.3% and 89.8% of the overall $ZnSe\leftrightarrow Se\leftrightarrow SeCl_4$ conversion, respectively.

We also estimated the specific energy of our Zn||Se cell based on the overall cell reaction as shown in Eq. (3), and compared it with representative AZB cells. As depicted in Fig. 5c, the maximum specific energy of our Zn||Se cell reached 404.2 Wh $kg^{-1}$, considerably outperforming recently reported AZBs based on intercalation-type cathodes like Zn||α-MnO$_2$ (217.9 Wh $kg^{-1}$)[43], Zn||β-MnO$_2$ (254.0 Wh $kg^{-1}$)[44], Zn||V$_2$O$_5$ (144.0 Wh $kg^{-1}$)[45], Zn||VO$_2$ (213.0 Wh $kg^{-1}$)[46], and Zn||calix[4]quinone (220.0 Wh $kg^{-1}$)[47]; as well as AZBs based on conversion-type cathodes like Zn||Te (287.9 Wh $kg^{-1}$)[40], Zn||Se (63.4 Wh $kg^{-1}$)[21], and Zn||S (238.1 Wh $kg^{-1}$)[18]. These results underscore the potential of our Zn||Se configuration for fabricating AZB devices with high energy densities. In addition, the rate performance of the Zn||Se cell was assessed with GCD measurements at varying specific currents (Supplementary Fig. 29 and 30). Congruous GCD shapes were preserved from 0.5 to 1.0 A $g_{Te}^{-1}$, achieving a specific capacity of 887.2 mAh $g_{Se}^{-1}$ at 1.0 A $g_{Se}^{-1}$.

Importantly, ZCE-Br enabled the Zn||Se cell to maintain specific capacities of 1246.8 mA $g_{Se}^{-1}$ after 50 cycles and 888.2 mA $g_{Se}^{-1}$ after 200 cycles, respectively (Fig. 5d). This result starkly contrasts with the rapid capacity decay observed for the Zn||Se cell with ZCE, which dropped to 394.1 mAh $g_{Se}^{-1}$ after 50 cycles. The comparison of GCD profiles for the Zn||Se cell with ZCE and ZCE-Br at 3rd, 10th, 20th and 30th is shown in Supplementary Fig. 31, further highlighting the enhanced stability by the ZCE-Br electrolyte. In addition, we prepared a Se cathode (denoted H-Se) with a high Se/AC ratio of 3.1 to further evaluate the effect of ZCE-Br in stabilizing its performance (Supplementary Fig. 32). In ZCE, the specific capacity of the H-Se electrode underwent a sharp decline from 1565.0 mAh $g_{Se}^{-1}$ to 529.2 mAh $g_{Se}^{-1}$ and 160.8 mAh $g_{Se}^{-1}$ after 10 cycles and 20 cycles at 0.2 mA $g_{Se}^{-1}$, respectively (Supplementary Fig. 33). In contrast, ZCE-Br enabled the H-Se electrode to retain 704.4 mAh $g_{Se}^{-1}$ after 100 cycles at 0.2 mA $g_{Se}^{-1}$.

## Discussion
In summary, our study has demonstrated the reversible six-electron $Se^{2-}/Se^0/Se^{4+}$ conversion via an overall $ZnSe\leftrightarrow Se\leftrightarrow SeCl_4$ process, with the $Br^-/Br_n^-$ redox couple effectively stabilizing the Se conversion. In the ZCE electrolyte, we observed rapid capacity decay of the Se cathode undergoing six-electron conversion, primarily attributed to the dissolution of SeCl$_4$ into the electrolyte and subsequent migration to the Zn anode, leading to dead Se passivation on the Zn anode. This shuttle effect resulted in continuous loss of active material in the Se cathode and fast capacity decay (from 1937.3 to 394.1 mAh $g_{Se}^{-1}$ after 50 cycles). By introducing bromide salt as an electrolyte additive, the $Br^-/Br_n^-$ redox couple was incorporated into the Zn||Se cell. Significantly, the generated $Br_n^-$ species acted as the dead-Se revitalizer, reacting with Se to scavenge dead Se from the Zn anode and regenerate active Se for cathode conversion. Consequently, the cycling stability of the Se electrode was considerably improved with high

capacities of 1246.8 and 888.2 mAh $g_{Se}^{-1}$ maintained after 50 and 200 cycles, respectively. Moreover, the demonstrated Zn||Se cell exhibited a maximum specific capacity of 2077.6 mAh $g_{Se}^{-1}$ and a maximum specific energy of 404.2 Wh $kg^{-1}$ based on the overall cell reaction. We believe that the Zn||Se cell, relying on the reversible $Se^{2-}/Se^{0}/Se^{4+}$ conversion, represents a promising AZB configuration for high-energy purpose. Besides, the concept of dead-material revitalizer holds promise for stabilizing conversion-type electrodes in different types of rechargeable batteries. In addition, we acknowledge the room for further improvement in the performance of the Se cathode, particularly in terms of active material ratio, loading mass, rate capability, and cycling performance. Addressing these aspects will require significant future efforts devoted to system optimization, such as developing advanced porous hosts with strong confinement effects and employing coated interphase/functionalized separators to inhibit the dissolution of active Se species.

## Methods

### Chemicals
Selenium (Se, powder, ≥99.5%), selenium tetrachloride ($SeCl_4$, powder), zinc selenide (ZnSe, powder, 99.99%), zinc chloride ($ZnCl_2$, ≥98%), lithium bromide (LiBr, anhydrous, ≥99%), zinc bromide ($ZnBr_2$, anhydrous), tetraethylammonium bromide ($Et_4NBr$, 98%), poly(vinylidene fluoride) (PVDF; weight-average molecule weight, $M_w \approx 275,000$), poly(ethylene oxide) (PEO, viscosity-average molecular weight, $M_v \approx 600,000$), glass fiber membrane (Grade GF/C, 0.2 mm thickness, 0.7 μm pore size), super P, N-methyl-2-pyrrolidone (NMP, ≥99.8%) and Zn foil were purchased from Sigma Aldrich. Activated carbon (AC) was purchased from Kuraray (Japan, YP80). Graphene oxide (GO) was purchased from GaoxiTech Co., Ltd. Carbon cloth (thickness: ~250 μm) was purchased from The Fuel Cell Store. All chemicals were directly used without further purification.

### Preparation of the Se electrode
Se-AC composite was first prepared by a melt-diffusion method. Specifically, 0.6 g of selenium and 1.4 g of AC were mixed homogeneously and loaded in a quartz ampoule and heat-sealed under vacuum condition (<$10^{-5}$ bar). For the preparation of H-Se, 1.6 g of selenium and 0.4 g of AC were used. We selected commercial YP80 AC as the standard host material for Se. YP80 is a commercially available product supplied by Kuraray. It features a microporous structure with a primary pore size of ~1 nm, a specific surface area of 2271 $m^2 g^{-1}$, and a bulk density of 0.18 g $ml^{-1}$ which provide ample space to accommodate active Se species. Additionally, the conductive nature of AC facilitates efficient charge transport, ensuring high utilization efficiency of the active material. Subsequently, the ampoule was heated to 600 °C at a rate of 5 °C $min^{-1}$ and maintained at this temperature for 12 h. The resulting Se-AC composite was collected and stored in a glovebox (under argon atmosphere with ≤0.1 ppm $O_2/H_2O$) for further experiments. To prepare the Se cathode, we dispersed Se-AC composite, super P, and PVDF (binder) in NMP with a mass ratio of 8:1:1 and mixed them in a mortar with a pestle thoroughly to achieve homogeneity in air atmosphere. Approximately 50 mL of NMP was added per gram of PVDF. The mortar and pestle used were made of agate. The resulting mixture was then coated onto a carbon cloth substrate by drop casting method and dried under vacuum at 80 °C for 12 h. The mass loading of Se was 1 - 1.5 mg $cm^{-2}$.

### Preparation of the hydrogel electrolyte
The glass fiber separator (Whatman GF/C, 10 mm in diameter; 0.2 mm in thickness) was first coated with graphene oxide (GO) through vacuum filtration. The GO dispersion (0.4 mg $ml^{-1}$) was prepared by dispersing GO into deionized water. After sonication, the 10 mL of the GO dispersion was filtered through a round-shaped glass fiber separator (47 mm in diameter). The GO-decorated separators were

then dried in a vacuum oven at 80 °C for 12 h. The loading amount of GO was around 0.25 mg $cm^{-2}$. Next, the separator was immersed in a hydrogel composed of 30 m $ZnCl_2$, 0.1 m $Et_4NBr$, and 10 wt% PEO at 80 °C overnight. Afterward, the excess hydrogel on the separator surface was removed, and the separator was dried in an oven at 80 °C for 6 h. For electrochemical measurements with ZCE, the separator was treated in a similar way, except using a hydrogel composed of 30 m $ZnCl_2$ and 10 wt% PEO.

### Electrochemical measurements
Titanium rods (99.5% purity with base area of 0.79 $cm^{-2}$) and stainless-steel rods (type SS304 with base area of 0.79 $cm^{-2}$) were used as current collectors for cathode and anode, respectively. Before use, the current collectors were cleaned with deionized water and ethanol. Electrochemical measurements were conducted using Swagelok cases (PFA-820-6, purchased from Swagelok Company) with Zn foil (diameter: 8 mm; thickness: 280 μm; mass: 86 mg) as the anode. A stainless-steel spring (type SS304 with force constant of 4.2 N $mm^{-1}$) was used in the Swagelok cell to secure the electrode on the current collectors.

CV curves were recorded using a CHI760E electrochemical workstation (CH Instruments), while GCD curves were collected with a Land battery test system (LAND CT2001A) at 25 °C. The GCD curves were used to calculate the specific energy (E) based on Eq. (7), where I represents the applied specific current based on the mass of Se, t represents the discharge duration, and U corresponds to the cell output voltage. The average voltage was subsequently calculated with Eq. (8), where E represents specific energy and Q denotes the specific capacity.

$$E = I \int_0^t U(t)dt \qquad (7)$$

$$\bar{U} = E/Q \qquad (8)$$

### Characterization
The material composition and morphology were analyzed using multiple techniques. Thermogravimetric analysis (TGA) was conducted with a Netzsch DSC-204 F1 system, while crystalline structure was examined via X-ray diffraction (XRD, Bruker D8) with Cu Kα radiation ($\lambda$ = 0.154 nm). Microstructural and elemental analyses were performed using a Carl Zeiss Gemini 500 field-emission scanning electron microscope (SEM) equipped with an Oxford X-$max^N$ 150 energy dispersive X-ray spectrometer (EDX). Structural and compositional analyses were conducted using X-ray photoelectron spectroscopy (XPS, PHI-5702, Al $K_\alpha$ X-ray, 1486.6 eV), Raman spectroscopy (Bruker RFS 100/S spectrometer at wavelength of 532 nm) and time-of-flight secondary ion mass spectrometry (TOF-SIMS 5 with a bismuth ion source, ION-TOF, GmbH, Germany.) In-situ X-ray absorption spectroscopy (XAS) measurements were performed at beamline BM23 at European Synchrotron Radiation Facility (ESRF) and beamline P65 from Deutsches Elektronen-Synchrotron (DESY). The measurements were conducted at K edge of Se (12.5 - 12.8 keV). A transmission mode under room temperature was applied to obtain the data. A standard Se reference sample (diluted to 20 wt%) was placed after the transmission chamber for energy calibration. The first-derivative point of XANES was used to calibrate the energy for each measurement. Demeter software package was used to process and analyze the XAS data[48]. Athena was used to calibrate, normalize, and align the XANES and EXAFS spectra[48,49]. Wavelet-transformed Te K-edge EXAFS analysis was performed using the HAMA code, which was developed by H. Funke and M. Chukalina[50]. For in-situ measurements, modified 2025-type coin cells were employed, featuring a central aperture sealed with Kapton tape to permit X-ray penetration. To mitigate corrosion, a

perforated titanium foil was inserted between the coin cell casing and the cathode.

## Data availability

The authors declare that all the relevant data are available within the paper and its Supplementary Information file or from the corresponding authors upon request. Source data are provided with this paper.

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

## Acknowledgements

This work was financially supported by European Union's Horizon 2020 research and innovation programme (LIGHT-CAP 101017821), and Deutsche Forschungsgemeinschaft (DFG, German Research Foundation), CRC1415 (Grant No. 417590517). J. D., J. Z., D. L., and X. C. thank China Scholarship Council for financial support. V.M. was supported by project LUAUS23049 from Ministry of Education Youth and Sports (MEYS). Z.S. was supported by ERC-CZ program (project LL2101) from Ministry of Education Youth and Sports (MEYS) and by the project Advanced Functional Nanorobots (reg. No. CZ.02.1.01/0.0/0.0/15_003/0000444 financed by the ERDF). The authors acknowledge the use of the facilities in the Dresden Center for Nanoanalysis (DCN) at Technische Universität Dresden, the GWK support for providing computing time through the Center for Information Services and High-Performance Computing (ZIH) at TU Dresden, and beam time allocation at beamline P65 at the PETRA III synchrotron (DESY, Hamburg, Germany) and beamline BM23 at ESRF synchrotron (Grenoble, France).

## Author contributions

J.D., X.F., and M.Y. conceived the idea and designed the experiments. J.D. synthesized the electrode materials and electrolytes. J.Z. and X.C. performed the Raman study. H.X. performed TOF-SIMS under the supervision of J.X.Z. Y.Z., V.M., and Z.S. performed XPS. M.L. and D.L performed the SEM. J.D., G.W., and Jie D. performed the electrochemical measurements. J.D., X.C., Q.G., A.M. and D.M. performed the XAS study. J.D. wrote the manuscript under the supervision of X.F. and M.Y., and all authors read and commented on the manuscript.

## Funding

## Competing interests

The authors declare no competing interests.
