## [Transparent Peer Review file · Nature Communications]

Six-electron-conversion selenium cathodes stabilized by dead-selenium revitalizer for aqueous zinc batteries

Corresponding Author: Professor Xinliang Feng

Version 0:

Reviewer comments:

Reviewer #1

(Remarks to the Author)

The paper reports a Zn/SeCl_xBr_y cell (x+y=4). Small amount of Br was added in ZnCl₂ electrolyte as a redox mediator to prolong the cycle life. I suggest the paper to be reconsidered after the the authors try to address the following comments:

1. The charge product needs to be carefully determined and confirmed, because it is critical for the calculation of specific energy. If it's SeCl₄, the energy is above 400 Wh/kg as shown in Figure 5c, but if it's SeBr₄, the energy will drop below 300 Wh/kg.
2. In Figure 2c, the authors showed some XPS results, they should also show the survey spectra and elemental ratio between Cl and Br to determine what is the charge product. The sample has to be washed after being harvested from cycled cell.
3. SeCl₄ will react with water to form H₂SeO₃. Is this happening in their electrolyte? The authors should provide evidence.
4. In the cathode, the amount of activated carbon is more than double the weight of Se. In addition, the authors also introduce another conductive carbon. Is this because the conductivity of Se is insufficient? Further explanation is needed.
5. The study investigates the capacity decay of Zn in aqueous electrolytes without poly(ethylene oxide), followed by the introduction of bromide salts, which are tested in a hydrogel electrolyte. This comparison is inconsistent and not parallel. Please provide detailed morphology and composition studies of the zinc anode cycled in the actual ZCE electrolyte (composed of 30M ZnCl₂ and poly(ethylene oxide)).
6. If the polybromide can oxidize Se, it more readily oxidizes Zn anode and cause self-discharge and shape change. Please provide evidence-supported explanation.
7. The paper lacks performance testing of the zinc anode in the modified electrolyte, including assessments of coulombic efficiency and cycling stability. These are crucial metrics that should be provided.
8. To conclusively demonstrate that the introduction of bromide salts removes the passivation layer on the zinc anode, it is suggested that the authors provide more convincing evidence through additional XRD, XPS, and Raman analyses, consistent with the earlier experiments.
9. The manuscript lacks crucial experimental details, such as the amount of electrolyte added and the thickness and mass of the zinc foil used. These should be included to ensure the reproducibility of the experiments.
10. The authors calculate performance metrics based on the total mass of active materials in both electrodes, whereas some referenced studies calculate metrics based on the total mass of the cathode and anode. This inconsistency in standards renders the performance comparison unreasonable. The authors should adopt a uniform standard for all performance comparisons.

Reviewer #2

(Remarks to the Author)

The authors reported a reversible six-electron-conversion Se cathode undergoing a ZnSe/Se/SeCl₄ reaction and introduced a Br⁻/Br^{m-} redox couple to activate the dead-selenium. Although the Br⁻/Br^{m-} redox couple partially improved cycling performance, the Zn//Se cell with ZCE-Br still experienced rapid capacity, maintaining only a ~43% capacity retention rate after 200 cycles at 0.5A/g. This performance is less impressive compared to previously reported aqueous Zn//Se batteries (doi.org/10.1002/aenm.202201322). Besides, the initiation of six-electron-conversion of chalcogens in ZnCl₂-based electrolyte has been previously documented (10.1021/jacs.3c06488), and a similar electrolyte recipe has been reported by the authors in a recent report (10.1002/adma.202313621), which largely reduces the novelty of this work. In addition, the

function mechanism of the proposed ZCE-Br electrolyte and the failure mechanism of the battery in this electrolyte remain largely unexplored. Therefore, I would recommend the authors to provide more insightful information and submit it to a more specialized journal. Specific concerns are as follows :

1. The weight ratio of active materials significantly impacts the electrochemical performance of Zn//Se cells, such as energy density, cycle stability, and rate capability. In this study, the Se weight ratio is ~22%, which is far lower than reported in previous works. The authors should increase the loading to an average level of 50% to facilitate accurate performance evaluation and comparison with previous studies.
2. Please elaborate on the thermogravimetric curve and provide the calculation details for the final selenium and activated carbon ratio. The atmosphere used for the thermogravimetric test should also be given.
3. This manuscript mentions that the ZCE-Br electrolyte improves the cycle life of Zn//Se batteries, and no Se signal was detected at the zinc anode after 200 cycles. However, the capacity of the Se cathode still declines rapidly during cycling. What is the failure mechanism of this battery system?
4. Fig. S9 shows that the cycling performance of the Se electrode in ZCE with different bromide salts varies depending on the cations. How do the cations affect the stability of the Zn//Se cells?
5. How does the concentration of the Br⁻/Brn⁻ redox couple affect the electrochemical performance of Zn//Se cells?
6. While the Br⁻/Brn⁻ redox couple can rejuvenate dead-selenium, the crossover of Br species may corrode the Zn anode, posing potential issues for the Zn//Se system. Therefore, the electrochemical performance of the Zn anode in the ZCE-Br electrolyte should be systematically studied.
7. Br species usually exhibit poor reaction kinetics in common electrolytes, thus the reversibility of Br⁻/Brn⁻ redox couple should be discussed in this work.
8. Please provide the ionic conductivity of the ZCE and ZCE-Br electrolytes.
9. The initial GCD profile and the initial Coulomb efficiency of the Zn//Se cell in both the ZCE-Br and ZCE-Br electrolyte should be provided and discussed.
10. As shown in Supplementary Fig. 19, the polarization of the Zn//Se cell in the ZCE-Br electrolyte increases with cycling, why?
11. Supplementary Table 1 should include electrode areal load and active material/host weight ratio to better understand the overall performance. Detailed information on other conversion-type cathodes (Zn-S, Zn-Se, Zn-Te et al.) is also necessary for comparison.

Reviewer #3

(Remarks to the Author)

This study presents an innovative six-electron Se conversion cathode chemistry for aqueous zinc batteries. The demonstrated Zn//Se cells show remarkable capacity and energy density, making a significant contribution to the field. Moreover, the introduction of a "dead-selenium revitalizer" strategy is particularly noteworthy, as it offers a solution to the challenge of capacity decay due to shuttling issues, which could have broader implications for other battery systems facing similar challenges. The manuscript is well-written, and the data comprehensively supports the conclusions. This work is suggested for publication in Nature Communications, pending the authors' attention to the following issues.

1. The claim and comparison of energy density should be rigorous and objective. The authors are suggested to clearly state their calculation method for their energy density and confirm all values of listed energy density are calculated using the same standard. An estimated energy density based on the total mass of the cell is more convincing.
2. Since the bromide redox couple plays a critical role as the inactive "dead" Se revitalizer to stabilize the electrode performance. I wonder if this is also applicable to aqueous sulfur batteries or if iodide redox serves a similar function. If this strategy applies to various multi-electron conversion-type cathodes, this work would be more impactful.
3. In the XPS data of the pristine Se⁰ state, there are distinct peaks labeled Se-C that show higher binding energy compared to Se⁰. The authors should provide a more detailed explanation for why these peaks appear in the pristine sample.
4. The XPS data suggests a possible phase change after Se melting in AC. To provide a more comprehensive characterization, please include XRD patterns of both the pristine Se and the Se-AC composite.
5. The preparation process of the quasi-solid electrolyte is not clearly explained, which is crucial for readers to replicate your experiments. The authors are suggested to include some optical images that illustrate the steps involved in preparing the quasi-solid electrolyte.
6. Why was YP80 AC chosen as the host material? Did the author explore other host materials for Se? What impact do different host materials have on improving the stability of the cells? These issues should be further clarified in the manuscript.
7. In Fig. 5a, the discharge capacity of the Zn//Se cell is indicated as 2077 mAh/g. The authors estimate that after subtracting the capacity contribution of AC, a capacity of 1907.6 mAh/g is attributed to the Se conversion reaction. However, there is likely some contribution from the Br⁰/Br⁻ conversion as well. The authors should account for this contribution to determine the true capacity of the Se conversion reaction more accurately.

Version 1:

Reviewer comments:

Reviewer #1

(Remarks to the Author)

My previous comments have been addressed by the authors. I recommend publication of this revised manuscript.

Reviewer #2

(Remarks to the Author)

After carefully reconsidering the manuscript based on the authors' responses to my review comments, I find that, while the authors have made efforts to address the concerns raised, the manuscript does not fully meet the rigorous standards expected for publication in a decent journal like Nature Communications. Firstly, recent literature has reported that superhalide-anion-containing electrolytes can activate the six-electron conversion of selenium (10.1021/acs.nanolett.4c00198), indicating that the $\text{Se}^{2-}/\text{Se}^0/\text{Se}^{4+}$ process is not novel. Although the Zn//Se battery achieved higher capacities compared to previous systems, this improvement appears to be incremental rather than a fundamental breakthrough. Secondly, the concept of a dead-material revitalizer, mentioned by the authors, is not new and has been reported in other types of batteries (10.1038/s41560-021-00789-7; 10.1002/anie.202110589). Thirdly, the long-term cycling stability of the Zn//Se battery in this work falls short of previously reported aqueous Zn//Se batteries (10.1039/D0EE02999H; 10.1002/adv.202403224; 10.1002/aenm.202201322). Furthermore, the failure mechanism of the system and the long-term stability of the Zn//Se battery require more extensive exploration.

Given these considerations, while the manuscript contains valuable insights, I believe it would be more suitable for submission to a more specialized journal where these issues can be explored in greater depth. Additionally, there are a few minor issues in the paper that require clarification:

1. The Se-C bond is present in the pristine sample in Figure 3c, but it disappears during the charge/discharge process. Is the Se-C bond converted during this process, and if so, how does this affect the electrochemical performance?
2. Both Supplementary Figure 4a and Supplementary Figure 16b should depict the charge/discharge curves of Zn//AC batteries, but the curves in these two figures differ, with a discharge platform clearly visible at 1.4 V in Supplementary Figure 16b.

Reviewer #3

(Remarks to the Author)

The authors have done considerable work to address the reviewers' concerns. The revised manuscript is acceptable for publication in Nat Commun. I have only minor further revision comments to improve the academic rigor of this paper. The authors are suggested to report the potential of the positive electrodes, not voltage, as appropriate electrochemistry terminology, as Nat. Nanotechnol. called upon recently (<https://www.nature.com/articles/s41565-024-01844-6>). Additionally, the reference redox couple should be "Zn²⁺/Zn", not just "Zn".

To Reviewer 1:

The paper reports a Zn/SeCl_xBr_y cell (x+y=4). Small amount of Br was added in ZnCl₂ electrolyte as a redox mediator to prolong the cycle life. I suggest the paper to be reconsidered after the authors try to address the following comments:

Response: We appreciate the constructive comments from the reviewer, which are invaluable in improving the quality of our manuscript. Accordingly, we have conducted additional experiments and discussions in the revised manuscript. Detailed point-by-point responses are provided below.

1. The charge product needs to be carefully determined and confirmed, because it is critical for the calculation of specific energy. If it's SeCl₄, the energy is above 400 Wh/kg as shown in Fig. 5c, but if it's SeBr₄, the energy will drop below 300 Wh/kg.

Response: Thank you for the constructive comment. We believe that SeCl₄ is the predominant charge product, while the generation of SeBr₄ or Se(Br_n)₄ could constitute a minor fraction of the charge product. This statement is based on the fact that the Cl⁻ concentration is approximately 300 times higher than the Br⁻ concentration in the employed ZCE-Br electrolyte. It is also supported by the Raman spectrum of the fully charged Se electrode in a-ZCE-Br (**Fig. R1**). The spectrum exhibits characteristic peaks of SeCl₄ at 126 cm⁻¹, 253 cm⁻¹, and a broaden peak ranging from 474 cm⁻¹ to 515 cm⁻¹.¹ Meanwhile, peaks associated with SeBr₄ are barely detectable.² The corresponding discussion has been added to the revised manuscript (**Page 12, Paragraph 1**).

Fig. R1 Raman spectrum of the fully charged Se electrode in a-ZCE-Br.

2. In Fig. 2c, the authors showed some XPS results, they should also show the survey spectra and elemental ratio between Cl and Br to determine what is the charge product. The sample has to be washed after being harvested from cycled cell.

Response: According to your suggestion, the XPS survey spectrum of the fully charged Se electrode

after washing is provided in **Fig. R2a**. Pronounced Se and Cl signals are observed, whereas the Br signal is not visible. The same conclusion can be derived from the high-resolution Se 3d (**Fig. R2b**), Cl 2p (**Fig. R2c**), and Br 3d (**Fig. R2d**) XPS spectra. These findings further support SeCl₄ as the predominant charge product rather than SeBr₄. The corresponding discussion has been added to the revised manuscript (**Page 12, Paragraph 1**).

Fig. R2 a The survey, **b** Se 3d, **c** Cl 2p, and **d** Br 3d XPS spectra of the fully charged Se electrode in a-ZCE-Br.

3. SeCl₄ will react with water to form H₂SeO₃. Is this happening in their electrolyte? The authors should provide evidence.

Response: Thank you for the insightful question. To assess the potential hydrolysis of SeCl₄ in our electrolyte, we directly added SeCl₄ to 30 m ZnCl₂, the base electrolyte for our study. The resulting solution was then analyzed by Raman spectroscopy. As shown in **Fig. R3a**, the sample presents the characteristic SeCl₄ peaks at around 163 cm⁻¹ and 375 cm⁻¹,¹ along with the peak associated with the [ZnCl₄]²⁻ species³ and the background peak of the glass vessel (**Fig. R3b**). No peaks associated with H₂SeO₃ are detectable⁴. This result indicates that the hydrolysis of SeCl₄ to form H₂SeO₃ is suppressed in the highly concentrated ZnCl₂ electrolyte due to the reduced reactivity of water. The corresponding discussion has been added to the revised manuscript (**Page 10, Paragraph 1**).

Fig. R3 Raman spectra of **a** 5 m SeCl_4 dissolved in 30 m ZnCl_2 and **b** the glass vessel used to contain the solution.

4. In the cathode, the amount of activated carbon is more than double the weight of Se. In addition, the authors also introduce another conductive carbon. Is this because the conductivity of Se is insufficient? Further explanation is needed.

Response: We appreciate the constructive question from the reviewer. In this study, we utilized a large amount of activated carbon in the fabrication of the Se electrode primarily because its porous structure serves as an effective host for active Se-species confinement. This design mitigates dissolution and prevents structural collapse. Specifically, the Se/SeCl_4 conversion process undergoes a considerable volume expansion of over 500%, presenting a critical challenge to achieving electrochemical stability. Additionally, the conductivity of the Se electrode is a concern, as the discharge product (i.e., ZnSe) is an intrinsic semiconductor with large bandgap⁵.

We acknowledge that the use of a large amount of inactive carbon can reduce the practicality of the obtained electrode. To address this, we further prepared a Se-AC compound with a large Se weight ratio of 75.5% (**Fig. R4a**). This composite was then employed to fabricate a Se electrode (denoted H-Se). In ZCE-Br, the H-Se electrode delivered an initial specific capacity of 1602.8 $\text{mAh g}_{\text{Se}}^{-1}$ (**Fig. R4b**), corresponding to an energy density of 1823 $\text{Wh kg}_{\text{Se}}^{-1}$. Moreover, the significant role of $\text{Br}^-/\text{Br}_n^-$ redox couple in stabilizing the $\text{ZnSe}/\text{Se}/\text{SeCl}_4$ conversion was also demonstrated using the H-Se electrode. In ZCE, the specific capacity of the H-Se electrode underwent a sharp decline from 1565.0 $\text{mAh g}_{\text{Se}}^{-1}$ to 529.2 $\text{mAh g}_{\text{Se}}^{-1}$ and 160.8 $\text{mAh g}_{\text{Se}}^{-1}$ after 10 cycles (**Fig. R4c**) and 20 cycles (**Fig. R4d**) at 0.2 $\text{mA g}_{\text{Se}}^{-1}$, respectively. In contrast, ZCE-Br enabled the H-Se electrode to retain 704.4 $\text{mAh g}_{\text{Se}}^{-1}$ after 100 cycles at 0.2 $\text{mA g}_{\text{Se}}^{-1}$ (**Fig. R4d**). The corresponding discussion has been added to the revised manuscript (**Page 16, Paragraph 2**).

Fig. R4 a Thermogravimetric analysis of the Se-AC compound with a high Se weight ratio of 75.5%. The initial 10-cycle GCD profiles of the H-Se electrode in **b** ZCE-Br and **c** ZCE at 0.2 A g_{Se}⁻¹. **d** Cycling performance of the H-Se electrode in ZCE and ZCE-Br at 0.2 A g_{Se}⁻¹.

5. The study investigates the capacity decay of Zn in aqueous electrolytes without poly(ethylene oxide), followed by the introduction of bromide salts, which are tested in a hydrogel electrolyte. This comparison is inconsistent and not parallel. Please provide detailed morphology and composition studies of the zinc anode cycled in the actual ZCE electrolyte (composed of 30 M ZnCl₂ and poly(ethylene oxide)).

Response: We are sorry for the confusion raised in our discussions. In fact, we carried out analyses of the Zn anode in electrolytes both without (i.e., a-ZCE) and with (i.e., ZCE and ZCE-Br) poly(ethylene oxide). Initially, we focused on a-ZCE to confirm the issues of Se-species shuttling and dead Se formation without the complications that poly(ethylene oxide) might introduce during characterization (Fig. 3a-e). Subsequently, we compared the SEM images and EDX spectra of Zn anodes after cycling in both ZCE and ZCE-Br (Fig. 3f-i or Fig. R5). Compared with the cycled Zn anode in ZCE (Fig. R5a), the cycled Zn anode in ZCE-Br exhibited a notably smoother surface (Fig. R5b). Moreover, while a distinct Se signal was detected on the cycled Zn anode in ZCE (Fig. R5c), almost no Se signals were observed on the cycled Zn anode in ZCE-Br (Fig. R5d). These results evidence the successful suppression of dead Se formation in ZCE-Br.

Fig. R5 SEM images of the Zn anodes in Zn//Se cells with **a** ZCE and **b** ZCE-Br after 10 cycles at $0.5 \text{ A g}_{\text{Se}}^{-1}$. EDX spectra of the Zn anodes in Zn//Se cells with **c** ZCE and **d** ZCE-Br after 10 cycles at $0.5 \text{ A g}_{\text{Se}}^{-1}$.

To further address your concern, we performed additional Raman and XPS analyses on the cycled Zn anodes in both ZCE and ZCE-Br. As shown in **Fig. R6**, the Zn anode cycled in ZCE exhibits obvious Se signals, indicating the presence of dead Se. In contrast, the Se signal is barely detectable on the Zn anode cycled in ZCE-Br, further verifying the suppression of dead Se formation. The corresponding discussion has been added to the revised manuscript (**Page 10, Paragraph 3**).

Fig. R6 a Raman and **b** Se 3d XPS spectra of the Zn anodes in the Zn//Se cells with ZCE and ZCE-Br after 10 cycles at $0.5 \text{ A g}_{\text{Se}}^{-1}$.

6. If the polybromide can oxidize Se, it more readily oxidizes Zn anode and cause self-discharge and shape change. Please provide evidence-supported explanation.

Response: Indeed, polybromide species (Br_n^-) are known to react with Zn metal.⁶ To assess their effect on the full cell, we carried out a self-discharge test on the Zn//Se cells with ZCE and ZCE-Br (**Fig. R7**). Both cells were initially charged to 1.9 V at $0.2 \text{ A g}_{\text{Se}}^{-1}$, then kept under open-circuit standing for 24 h, and finally discharged to 0.05 V at $0.2 \text{ A g}_{\text{Se}}^{-1}$. As revealed, both cells retained their original discharge curve shapes after the open-circuit period. The Zn//Se cell with ZCE-Br exhibited a slightly lower

coulombic efficiency (79.4%) than the cell with ZCE (83.6%), likely due to the reaction of shuttled Br_n^- species with the Zn metal anode. Nevertheless, the effect of Br_n^- on the self-discharge issue is not significant, because only a limited amount of Br_n^- species are generated and can migrate to the anode during the discharge process of the Zn//Se cell. To further address this issue, surface coatings on Zn anode could be a viable approach to mitigate the reaction between Br_n^- and Zn metal, warranting further research. For example, a recent study demonstrated that an alginate-graphene oxide hydrogel coating on the Zn anode effectively protected it from corrosion by polybromide species.⁷ The corresponding discussion has been added to the revised manuscript (Page 10, Paragraph 2).

Fig. R7 Self-discharge test of the Zn//Se cells with **a** ZCE and **b** ZCE-Br. The cells were initially charged to 1.9 V at 0.2 A g_{Se}⁻¹, then kept under open-circuit standing for 24 h, and finally discharged to 0.05 V at 0.2 A g_{Se}⁻¹.

7. The paper lacks performance testing of the zinc anode in the modified electrolyte, including assessments of coulombic efficiency and cycling stability. These are crucial metrics that should be provided.

Response: We appreciate the constructive suggestion from the reviewer. Accordingly, we conducted a detailed analysis on the coulombic efficiency and cycling stability of the Zn anode in ZCE-Br. The coulombic efficiency was evaluated with the Zn//Cu asymmetric cell using the ‘Aurbach’ protocol⁸, wherein Zn was first deposited on Cu at 0.5 mA cm⁻² for 3 h, then the cell was cycled at 0.5 mA cm⁻² and 0.5 mAh cm⁻² for 30 cycles, and finally Zn was completely stripped from Cu. The coulombic efficiency was calculated by the total stripping capacity divided by the total plating capacity. As displayed in **Fig. R8a**, the Zn//Cu cell with ZCE-Br demonstrated a high coulombic efficiency of 98.8 %, which is only slightly lower than the Zn//Cu cell with ZCE. Furthermore, the cycling stability of the Zn//Zn symmetric cell with ZCE-Br was assessed at two different current densities. The cell exhibited

stable operation for 1000 hours at 1 mA cm^{-2} and 1 mAh cm^{-2} with a hysteresis voltage of less than 38 mV (**Fig. R8b**). Similarly, at 0.2 mA cm^{-2} and 0.2 mAh cm^{-2} , the cell maintained stable with a hysteresis voltage of less than 15 mV (**Fig. R8c**). The corresponding discussion has been added to the revised manuscript (**Page 11, Paragraph 1**).

Fig. R8 a Coulombic efficiencies (CEs) of Zn//Cu asymmetric cells with ZCE-Br and ZCE measured by the ‘Aurbach’ method. Voltage profiles of Zn//Zn symmetric cells with ZCE-Br at **b** 1 mA cm^{-2} under a fixed capacity of 1 mAh cm^{-2} and **c** 0.2 mA cm^{-2} under a fixed capacity of 0.2 mAh cm^{-2} .

8. To conclusively demonstrate that the introduction of bromide salts removes the passivation layer on the zinc anode, it is suggested that the authors provide more convincing evidence through additional XRD, XPS, and Raman analyses, consistent with the earlier experiments.

Response: Thank you for the valuable suggestion. We carried out additional XPS and Raman analyses on the cycled Zn anode in ZCE (**Fig. R6**), which fully support its surface passivation by Se. The detailed discussion can be found in our response to your Q5. In addition, we compared the XRD patterns of the cycled Zn anodes in ZCE and ZCE-Br (**Fig. R9**). No characteristic peaks associated with Se were

detected on both cycled Zn anodes, implying the low crystallinity or amorphous nature of dead Se on the Zn surface. The corresponding discussion has been added to the revised manuscript (**Page 10, Paragraph 3**).

Fig. R9 XRD spectra of the Zn anodes in Zn//Se cells with ZCE and ZCE-Br after 10 GCD cycles at $0.5 \text{ A g}_{\text{Se}}^{-1}$.

9. The manuscript lacks crucial experimental details, such as the amount of electrolyte added and the thickness and mass of the zinc foil used. These should be included to ensure the reproducibility of the experiments.

Response: Thank you for the kind reminder. The corresponding experimental details can be found in the revised manuscript or as below.

The glass fiber separator (Whatman GF/C, 10 mm in diameter; 0.2 mm in thickness) was first coated with graphene oxide (GO) with around 0.25 mg cm^{-2} through vacuum filtration. Next, the separator was immersed in a hydrogel composed of 30 m ZnCl_2 , 0.1 m Et_4NBr , and 10 wt% PEO at $80 \text{ }^\circ\text{C}$ overnight. Afterward, the excess hydrogel on the separator surface was removed, and the separator was dried in an oven at $80 \text{ }^\circ\text{C}$ for 6 h. For electrochemical measurements with ZCE, the separator was treated in a similar way, except using a hydrogel composed of 30 m ZnCl_2 and 10 wt% PEO.

Round-shape Zn foils with a diameter of 8 mm, a thickness of $280 \text{ }\mu\text{m}$, and a mass of 86 mg were used as the anodes for the electrochemical measurements. The corresponding discussion has been added to the revised manuscript (**Page 18, Paragraph 2-3**).

10. The authors calculate performance metrics based on the total mass of active materials in both electrodes, whereas some referenced studies calculate metrics based on the total mass of the cathode and anode. This inconsistency in standards renders the performance comparison unreasonable. The authors should adopt a uniform standard for all performance comparisons.

Response: Thank you for pointing out the inconsistency in our performance comparison. We made the necessary corrections to the relevant figures and table to ensure that all comparisons are based on the same standard. Specifically, the metrics shown in **Fig. R10a** (i.e., **Fig. 5b** in the manuscript) and **Table R1** (i.e., **Supplementary Table S1** in the manuscript) were all calculated based solely on the mass of active cathode materials. Additionally, the energy densities shown in **Fig. R10b** (i.e., **Fig. 5c** in the manuscript) were calculated based on the overall cell reaction, incorporating the masses of all active species, including the active anode material, cathode material, and any electrolyte salt involved in the electrochemical reaction.

Fig. R10 a Comparison of performance (encompassing gravimetric capacity, average discharge voltage, and gravimetric energy density) of six-electron Se conversion chemistry with reported chalcogen cathodes for AZBs. The specific capacities and energy densities are calculated based on the mass of active cathode materials. **b** Comparison of energy density between the Zn//Se device and recently reported AZB devices. The presented energy densities were calculated based on the overall cell reaction, incorporating the masses of all active species, including the active anode material, cathode material, and any electrolyte salt involved in the electrochemical reaction.

Table R1 Performance comparison with the reported AZB cathode materials.

Cathode materials	Electrolyte	Midpoint voltage (V vs. Zn)	Specific capacity (mAh g ⁻¹)	Energy density (Wh kg ⁻¹)	Ref.
Se (22.7% in Se-AC compound)	30 m ZnCl ₂ + 0.1 m Et ₄ NBr + PEO	~1.03	2077	2138	This work
Se (75.5% in Se-AC compound)	30 m ZnCl ₂ + 0.1 m Et ₄ NBr + PEO	~1.14	1603	1823	This work
α-MnO ₂	2 M ZnSO ₄ + 0.1 M MnSO ₄	~1.31	285	373	9
β-MnO ₂	3 M Zn(CF ₃ SO ₃) ₂ + 0.1 M Mn(CF ₃ SO ₃) ₂	~1.13	225	254	10
K _{0.8} Mn ₈ O ₁₆	2 M ZnSO ₄ + 0.1 M MnSO ₄	~1.24	320	398	11
γ-MnO ₂	1 M ZnSO ₄	~1.25	285	356	12
V ₂ O ₅	3 M Zn(CF ₃ SO ₃) ₂	~0.58	470	273	13
V ₃ O ₇ •H ₂ O	1 M ZnSO ₄	~0.64	375	240	14
Zn _{0.3} V ₂ O ₅ •1.5H ₂ O	3 M Zn(CF ₃ SO ₃) ₂	~0.79	426	337	15
Calix[4]quinone	3 M Zn(CF ₃ SO ₃) ₂	~1	335	335	16
Para-dinitrobenzene	3 M Zn(CF ₃ SO ₃) ₂	~0.57	402	230	17
Polyaniline	1 M Zn(CF ₃ SO ₃) ₂	~1.1	200	220	18
3, 7-bis(phenylamino)phenothiazin-5-ium iodide	2.0 M ZnSO ₄	~1.1	188	207	19
1,4,5,8,9,12-hexaazatriphenylene-based COFs.	2.0 M ZnSO ₄	~0.84	344	289	20

$\text{Na}_3\text{V}_2(\text{PO}_4)_3$	0.5 M $\text{Zn}(\text{CH}_3\text{COO})_2$	~1.1	97	107	21
$\text{FeFe}(\text{CN})_6$	1 M $\text{Zn}(\text{OAc})_2$	~1.3	122	159	22
$\text{CuFe}(\text{CN})_6$	20 mM ZnSO_4	~1.73	53	92	23
$\text{CoFe}(\text{CN})_6$	4 M $\text{Zn}(\text{CF}_3\text{SO}_3)_2$	~1.75	173.4	303	24
I_2	30 m ZnCl_2	~1.48	612.5	905	25
LF-PLSD	1 M $\text{Zn}(\text{TFSI})_2$	~0.63	1148	725	26
Se	4 M $\text{Zn}(\text{CF}_3\text{SO}_3)_2$ + PEO	~1.1	664.7	729	27
Se	1 M ZnSO_4	~1.23	611	751	28
TP-Se	1 M $\text{Zn}(\text{OTF})_2$	~1.95	72.9	142	29
Te	30 m ZnCl_2 + PEO	~0.84	1223.9	1028	30
Te	1 M ZnSO_4	~0.58	419	241	31
Te	30 m ZnCl_2	~0.83	802.7	~666	32

To Reviewer 2:

The authors reported a reversible six-electron-conversion Se cathode undergoing a ZnSe/Se/SeCl₄ reaction and introduced a Br⁻/Br_n⁻ redox couple to activate the dead-selenium. Although the Br⁻/Br_n⁻ redox couple partially improved cycling performance, the Zn//Se cell with ZCE-Br still experienced rapid capacity, maintaining only a ~43% capacity retention rate after 200 cycles at 0.5A/g. This performance is less impressive compared to previously reported aqueous Zn//Se batteries (doi.org/10.1002/aenm.202201322). Besides, the initiation of six-electron-conversion of chalcogens in ZnCl₂-based electrolyte has been previously documented (10.1021/jacs.3c06488), and a similar electrolyte recipe has been reported by the authors in a recent report (10.1002/adma.202313621), which largely reduces the novelty of this work. In addition, the function mechanism of the proposed ZCE-Br electrolyte and the failure mechanism of the battery in this electrolyte remain largely unexplored. Therefore, I would recommend the authors to provide more insightful information and submit it to a more specialized journal. Specific concerns are as follows:

Response: We appreciate the constructive comments from the reviewer, which provide valuable insights for improving the manuscript quality. Nevertheless, we politely disagree with the comment on the novelty and significance of our research. Actually, in this work, we demonstrate the reversible six-electron Se²⁻/Se⁰/Se⁴⁺ via a novel ZnSe/Se/SeCl₄ process, marking the first time that this specific process can be achieved. This conversion chemistry enables an ultrahigh specific capacity of 1937.3 mAh g_{Se}⁻¹, and the constructed Zn//Se cell achieves a maximum energy density of 404.2 Wh kg⁻¹ based on the overall cell reaction. We are grateful to the reviewer for pointing out previously reported Zn//Se batteries²⁷, in which Se⁰/Se⁴⁺ conversion is initiated in the presence of triflate anions (CF₃SO₃⁻). However, the large-size and multi-atom triflate anions introduce challenges related to conversion kinetics, resulting in practical issues like large charge/discharge polarization (0.54 V with electrocatalytic process and 1.07 V without electrocatalytic process) and low active material utilization efficiency (~32.6%). In contrast, our Se⁰/Se⁴⁺ conversion leverages the relatively small, single-atom Cl⁻ anions as charge carriers, enabling a low charge/discharge polarization (0.14 V) and high active material utilization efficiency (~85.7%). The reviewer also mentioned two latest studies on six-electron conversion of chalcogens in ZnCl₂-based electrolyte^{30,32}, including one from our group. However, both studies primarily evaluate Te conversion and, in our view, do not diminish the significance of our findings in Se conversion.

Another significant contribution of this study is the introduction of **a novel concept, namely the dead-material revitalizer, as a promising approach for stabilizing conversion-type battery electrodes.** Specifically, we identify the fast capacity decay of the Se electrode (from 1937.3 to 394.1 mAh g_{Se}⁻¹ after 50 cycles), which primarily originates from the dissolution of the charging product (*i.e.*, SeCl₄) into the electrolyte and the subsequent formation of dead Se on the Zn anode. Our work discloses that the incorporation of the Br⁻/Br_n⁻ redox couple into the Zn//Se cell enables the generated Br_n⁻ species to act as a dead-Se revitalizer. These species react with the Se passivation layer on the Zn anode, regenerating active Se for cathode reactions. As a result, the cycling stability of the Se electrode is significantly improved, with high capacities of 1246.8 and 888.2 mAh g_{Se}⁻¹ maintained after 50 and 200 cycles, respectively. Aligned with the positive views expressed by the other reviewers, we believe that the novelty and importance of this work justify its suitability for publication in *Nature Communications*. We agree with the reviewer that there is still room for further improvement in the performance of the Se cathode, especially its cycling stability. Fully addressing these remaining challenges will require extensive future efforts focused on system optimization, rather than one solution within a single research study. Significant inspiration can be drawn from the extensive research on non-aqueous rechargeable Li-S batteries, which date back to the 1960s and remain a highly active research area today. Nevertheless, we believe that the findings revealed in this study provide a valuable foundation and critical insights to guide future efforts in this direction. In addition, based on your comments, we have carefully revised the manuscript, including additional experiments and expanded discussions. The details are provided in the following point-by-point responses.

1. The weight ratio of active materials significantly impacts the electrochemical performance of Zn//Se cells, such as energy density, cycle stability, and rate capability. In this study, the Se weight ratio is ~22%, which is far lower than reported in previous works. The authors should increase the loading to an average level of 50% to facilitate accurate performance evaluation and comparison with previous studies.

Response: Thank you for the valuable advice. Accordingly, we prepared a Se-AC compound with a large Se weight ratio of 75.5% (**Fig. R11a**). This composite was then employed to fabricate a Se electrode (denoted H-Se). In ZCE-Br, the H-Se electrode delivered an initial specific capacity of 1602.8 mAh g_{Se}⁻¹ (**Fig. R11b**), corresponding to an energy density of 1823 Wh kg_{Se}⁻¹. These metrics have been

incorporated into **Fig. 5c** and **Supplementary Table 1** for comparison with previous studies. Moreover, the significant role of $\text{Br}^-/\text{Br}_n^-$ redox couple in stabilizing the $\text{ZnSe}/\text{Se}/\text{SeCl}_4$ conversion was also demonstrated using the H-Se electrode. In ZCE, the specific capacity of the H-Se electrode underwent a sharp decline from $1565.0 \text{ mAh g}_{\text{Se}}^{-1}$ to $529.2 \text{ mAh g}_{\text{Se}}^{-1}$ and $160.8 \text{ mAh g}_{\text{Se}}^{-1}$ after 10 cycles (**Fig. R11c**) and 20 cycles (**Fig. R11d**) at $0.2 \text{ mA g}_{\text{Se}}^{-1}$, respectively. In contrast, ZCE-Br enabled the H-Se electrode to retain $704.4 \text{ mAh g}_{\text{Se}}^{-1}$ after 100 cycles at $0.2 \text{ mA g}_{\text{Se}}^{-1}$ (**Fig. R11d**). The corresponding discussion has been added to the revised manuscript (**Page 16, Paragraph 2**).

Fig. R11 a Thermogravimetric analysis of the Se-AC compound with a high Se weight ratio of 75.5%. The initial 10-cycle GCD profiles of the H-Se electrode in **b** ZCE-Br and **c** ZCE at $0.2 \text{ A g}_{\text{Se}}^{-1}$. **d** Cycling performance of the H-Se electrode in ZCE and ZCE-Br at $0.2 \text{ A g}_{\text{Se}}^{-1}$.

2. Please elaborate on the thermogravimetric curve and provide the calculation details for the final selenium and activated carbon ratio. The atmosphere used for the thermogravimetric test should also be given.

Response: According to your suggestion, we have included detailed discussions on the thermogravimetric analysis (TGA) results in the revised manuscript. All TGA measurements were carried out under an argon atmosphere. As shown in **Fig. R12a**, the TGA profile of the Se-activated carbon (AC) compound with low Se content shows two mass loss steps, before $200 \text{ }^\circ\text{C}$ (loss of residual water confined in porous AC, 5.8%) and $550 \text{ }^\circ\text{C}$ (Se loss, 21.4%), respectively. The Se/AC ratio (R_L)

was thus estimated to be 0.3 according to **equation (R1)**. Furthermore, the TGA profile of the Se-AC compound with high Se content (**Fig. R12b**) shows only the Se loss step (75.5%), as the large Se content occupies the porous space of AC, leaving no room for water retention. The Se/AC ratio (R_H) was estimated to be 3.1 according to **equation (R2)**. The corresponding discussion has been added to the revised manuscript (**Page 5, Paragraph 2; Page 16, Paragraph 2**).

$$R_L = \frac{21.4\%}{1-21.4\%-5.8\%} \approx 0.3 \quad (\mathbf{R1})$$

$$R_H = \frac{75.5\%}{1-75.5\%} \approx 3.1 \quad (\mathbf{R2})$$

Fig. R12 TGA profiles of the Se-AC compound with **a** low Se content and **b** high Se content.

3. This manuscript mentions that the ZCE-Br electrolyte improves the cycle life of Zn//Se batteries, and no Se signal was detected at the zinc anode after 200 cycles. However, the capacity of the Se cathode still declines rapidly during cycling. What is the failure mechanism of this battery system?

Response: We appreciate the constructive questions from the reviewer. In this study, we identified two issues associated with the cycling stability of the Zn//Se cell, namely the dissolution of SeCl_4 and subsequent formation of dead Se on the Zn anode. The dead Se formation could accelerate the dissolution of SeCl_4 from the cathode, leading to a rapid loss of active cathode materials. The incorporation of the $\text{Br}^-/\text{Br}_n^-$ redox couple into the Zn//Se cell enables the generated Br_n^- species to act as a dead-Se revitalizer. These species react with the Se passivation layer on the Zn anode, regenerating active Se for cathode reactions. However, our strategy does not entirely prevent the dissolution of SeCl_4 from the cathode. Additionally, structural collapse in the cathode may contribute to the continuous capacity loss, particularly due to the substantial volume changes during the conversion reaction (e.g., >500% volume expansion from Se to SeCl_4 , > 160% volume expansion from Se to ZnSe). To

further address the issue, significant future efforts are desired to optimize the system, such as developing advanced porous hosts with strong confinement capabilities and employing coated interphase/functionalized separators to inhibit the dissolution of active Se species. The corresponding discussion has been added to the revised manuscript (**Supplementary Fig. 33**).

4. Fig. S9 shows that the cycling performance of the Se electrode in ZCE with different bromide salts varies depending on the cations. How do the cations affect the stability of the Zn//Se cells?

Response: Indeed, different bromide salts exhibited slightly varied effects in stabilizing the Se conversion reaction. However, the cation effect cannot be considered decisive, as all bromide salts demonstrated significant improvements in cycling performance. Among the three bromide salts tested, tetraethylammonium bromide (Et_4NBr) enabled the Zn//Se cell to achieve the best cycling performance. This can be attributed to the bulky nature of Et_4N^+ cations, which may associate with the conversion charging products to form large clusters^{33, 34}, thus mitigating its dissolution into the electrolyte. The corresponding discussion has been added to the revised manuscript (**Supplementary Fig. 11**).

5. How does the concentration of the $\text{Br}^-/\text{Br}_n^-$ redox couple affect the electrochemical performance of Zn//Se cells?

Response: Thank you for the valuable question. To address it, we prepared the ZCE-Br electrolytes with 0.1 m, 0.5 m, and 1 m Et_4NBr , which were further employed to assemble Zn//Se cells with the H-Se cathode. **Fig. R13** compares their GCD profiles at $0.2 \text{ A g}_{\text{Se}}^{-1}$. Increasing the Et_4NBr concentration results in a more pronounced discharge plateau at around 1.7 V, which corresponds to the $\text{Br}_n^-/\text{Br}^-$ reduction. This observation reflects that higher Et_4NBr concentrations trigger more $\text{Br}^-/\text{Br}_n^-$ redox reaction in the Zn//Se cell. However, higher Et_4NBr concentrations also lead to lower coulombic efficiency for the Zn//Se cell. That's because excess Br_n^- species generated during charging tend to migrate to the anode side and react with the Zn metal. Thereby, the Et_4NBr concentration must be optimized to minimize the generation of excess Br_n^- . With 0.1 m Et_4NBr , this negative effect could be restricted to minimal, enabling the Zn//Se cell to achieve a coulombic efficiency of 93.6% at the second cycle. The corresponding discussion has been added to the revised manuscript (**Page 10, Paragraph 2**).

Fig. R13 The 2nd GCD profiles of the Zn//Se cells with ZCE-Br containing **a** 0.1 m, **b** 0.5 m, and **c** 1 m Et₄NBr at 0.2 A g_{Se}⁻¹. The H-Se electrodes were used for the assembly of the Zn//Se cells.

6. While the Br⁻/Br_n⁻ redox couple can rejuvenate dead-selenium, the crossover of Br species may corrode the Zn anode, posing potential issues for the Zn//Se system. Therefore, the electrochemical performance of the Zn anode in the ZCE-Br electrolyte should be systematically studied.

Response: According to your valuable suggestion, we conducted a detailed analysis on the coulombic efficiency and cycling stability of the Zn anode in ZCE-Br. The coulombic efficiency was evaluated with the Zn//Cu asymmetric cell using the ‘Aurbach’ protocol⁸, wherein Zn was first deposited on Cu at 0.5 mA cm⁻² for 3 h, then the cell was cycled at 0.5 mA cm⁻² and 0.5 mAh cm⁻² for 30 cycles, and finally Zn was completely stripped from Cu. The coulombic efficiency was calculated by the total stripping capacity divided by the total plating capacity. As displayed in **Fig. R14a**, the Zn//Cu cell with ZCE-Br demonstrated a high coulombic efficiency of 98.8 %, which is only slightly lower than the Zn//Cu cell with ZCE. Furthermore, the cycling stability of the Zn//Zn symmetric cell with ZCE-Br was assessed at two different current densities. The cell exhibited stable operation for 1000 hours at 1 mA cm⁻² and 1 mAh cm⁻² with a hysteresis voltage of less than 38 mV (**Fig. R14b**). Similarly, at 0.2 mA cm⁻² and 0.2 mAh cm⁻², the cell maintained stable with a hysteresis voltage of less than 15 mV (**Fig. R14c**). The corresponding discussion has been added to the revised manuscript (**Page 11, Paragraph 1**).

Fig. R14 a Coulombic efficiencies (CEs) of Zn//Cu asymmetric cells with ZCE-Br and ZCE measured by the ‘Aurbach’ method. Voltage profiles of Zn//Zn symmetric cells with ZCE-Br at **b** 1 mA cm⁻² under a fixed capacity of 1 mAh cm⁻² and **c** 0.2 mA cm⁻² under a fixed capacity of 0.2 mAh cm⁻².

7. Br species usually exhibit poor reaction kinetics in common electrolytes, thus the reversibility of Br⁻/Br_n⁻ redox couple should be discussed in this work.

Response: To address your concern, we assembled Zn//AC cells with ZCE and ZCE-Br. The initial charge cycle of the Zn//AC cell with ZCE-Br exhibits a large irreversible capacity, which can be attributed to the oxidation of poly(ethylene oxide), generating ester species³⁵ at the cathode/electrolyte solid-solid interface (**Fig. R15a**). A similar irreversible capacity was also observed in the Zn//AC cell with ZCE (**Fig. R15b**). From the second cycle onward, the Zn//AC cell with ZCE-Br demonstrates almost identical GCD profiles, indicating the reversible Br⁻/Br_n⁻ conversion. Moreover, this good reversibility is evidenced by the superior cycling performance of the Zn//AC cell with ZCE-Br (**Fig. R15c**). The corresponding discussion has been added to the revised manuscript (**Page 11, Paragraph 1**).

Fig. R15 The initial three-cycle GCD profiles of the Zn//AC cell with **a** ZCE-Br and **b** ZCE at 0.2 A g⁻¹. **c** Cycling performance of the Zn//AC cell with ZCE-Br at 0.2 A g⁻¹.

8. Please provide the ionic conductivity of the ZCE and ZCE-Br electrolytes.

Response: According to your comment, we measured the ionic conductivities of ZCE and ZCE-Br using Swagelok cells with two stainless-steel rods as current collectors. Electrochemical impedance spectroscopy (EIS) was measured at 25 °C with an amplitude voltage of 5 mV over a frequency range of 100000 ~ 0.1 Hz. The ionic conductivities (σ) of ZCE and ZCE-Br were calculated to be 2.1×10^{-4} S cm⁻¹ and 3.1×10^{-4} S cm⁻¹, respectively, based on **equation (R3)**, where d (cm) represents the thickness of the electrolyte, S (cm²) is the electrolyte area, and R (Ω) is the resistance derived from the Nyquist plots (**Fig. R16**). The corresponding discussion has been added to the revised manuscript (**Page 11, Paragraph 1**).

$$\sigma = \frac{d}{SR} \quad (\text{R3})$$

Fig. R16 Nyquist plots of ZCE and ZCE-Br.

9. The initial GCD profile and the initial Coulomb efficiency of the Zn//Se cell in both the ZCE and ZCE-Br electrolyte should be provided and discussed.

Response: Thank you for the insightful suggestion. **Fig. R17a** displays the initial GCD profile of the Zn//Se cell with ZCE. In the initial charge cycle, the specific capacity (1732.9 mAh g_{se}⁻¹) significantly

exceeds the theoretical Se/Se⁴⁺ capacity (1358.7 mAh g_{Se}⁻¹). This additional capacity is attributed to the irreversible oxidation of poly(ethylene oxide) at the cathode/electrolyte solid-solid interface, as detailed in our response to your Q7. In the subsequent discharge cycle, the specific capacity (1937.3 mAh g_{Se}⁻¹) closely aligns with the theoretical Se²⁻/Se⁰/Se⁴⁺ conversion capacity (2038.0 mAh g_{Se}⁻¹), resulting in an initial coulombic efficiency of 111.8%. In the Zn//Se cell with ZCE-Br (**Fig. R17b**), both the initial charge (1946.2 mAh g_{Se}⁻¹) and discharge (2077.6 mAh g_{Se}⁻¹) capacities are slightly increased due to the incorporation of the Br⁻/Br_n⁻ redox couple. The initial coulombic efficiency (106.8%) is comparable to that of the Zn//Se cell with ZCE. The corresponding discussion has been added to the revised manuscript (**Page 15, Paragraph 2**).

Fig. R17 Initial GCD profiles of the Zn//Se cells with **a** ZCE and **b** ZCE-Br at 0.5 A g_{Se}⁻¹.

10. As shown in Supplementary Fig. 19, the polarization of the Zn//Se cell in the ZCE-Br electrolyte increases with cycling, why?

Response: We appreciate the constructive question. In fact, the increasing polarization observed during cycling is a common phenomenon for conversion-type electrodes, such as Cu₂S³⁶ and Si³⁷. This behavior can be assigned to structural degradation of the electrode caused by the large volume change that occur during charge and discharge. These volume changes lead to irreversible structural and morphological evolution, including particle pulverization and material breakdown. Such changes result in poor electrical contact, increased charge-transfer resistance, and, consequently, greater polarization. To address this issue, the development of advanced porous hosts with strong confinement capabilities, along with flexible binders possessing robust mechanical properties, is highly desirable and warrants extensive future research efforts. The corresponding discussion has been added to the revised manuscript (**Supplementary Fig. 31**).

11. Supplementary Table 1 should include electrode areal load and active material/host weight ratio to better understand the overall performance. Detailed information on other conversion-type cathodes (Zn-S, Zn-Se, Zn-Te et al.) is also necessary for comparison.

Response: According to your suggestions, the relevant information has been incorporated into **Supplementary Table 1** and can also be found in **Table R2**.

Table R2 Performance comparison with the reported AZB cathode materials.

Cathode materials	Mass load of active materials (mg cm^{-2})	Active materials/host ratio	Electrolyte	Midpoint voltage (V vs. Zn)	Specific capacity (mAh g^{-1})	Energy density (Wh kg^{-1})	Ref.
Se (22.7% in Se-AC compound)	1~1.5	0.3	30 m ZnCl_2 + 0.1 m Et_4NBr + PEO	~1.03	2077	2138	This work
Se (75.5% in Se-AC compound)	1~1.5	3.1	30 m ZnCl_2 + 0.1 m Et_4NBr + PEO	~1.14	1603	1823	This work
α - MnO_2	1~5	/	2 M ZnSO_4 + 0.1 M MnSO_4	~1.31	285	373	9
β - MnO_2	~2	/	3 M $\text{Zn}(\text{CF}_3\text{SO}_3)_2$ + 0.1 M $\text{Mn}(\text{CF}_3\text{SO}_3)_2$	~1.13	225	254	10
$\text{K}_{0.8}\text{Mn}_8\text{O}_{16}$	/	/	2 M ZnSO_4 + 0.1 M MnSO_4	~1.24	320	398	11
γ - MnO_2	/	/	1 M ZnSO_4	~1.25	285	356	12
V_2O_5	~2	4	3 M $\text{Zn}(\text{CF}_3\text{SO}_3)_2$	~0.58	470	273	13
$\text{V}_3\text{O}_7 \cdot \text{H}_2\text{O}$	/	/	1 M ZnSO_4	~0.64	375	240	14
$\text{Zn}_{0.3}\text{V}_2\text{O}_5 \cdot 1.5\text{H}_2\text{O}$	~2	/	3 M $\text{Zn}(\text{CF}_3\text{SO}_3)_2$	~0.79	426	337	15
Calix[4]quinone	2.5~10	/	3 M $\text{Zn}(\text{CF}_3\text{SO}_3)_2$	~1	335	335	16
Para-dinitrobenzene	1.8~2.1	1.5	3 M $\text{Zn}(\text{CF}_3\text{SO}_3)_2$	~0.57	402	230	17

Polyaniline	~1.5	/	1 M Zn(CF ₃ SO ₃) ₂	~1.1	200	220	18
3, 7-bis(phenylamino)phenothiazin-5-ium iodide	~4	/	2.0 M ZnSO ₄	~1.1	188	207	19
1,4,5,8,9,12-hexaazatriphenylene-based COFs.	1~1.5	/	2.0 M ZnSO ₄	~0.84	344	289	20
Na ₃ V ₂ (PO ₄) ₃	/	/	0.5 M Zn(CH ₃ COO) ₂	~1.1	97	107	21
FeFe(CN) ₆	~5	/	1 M Zn(OAc) ₂	~1.3	122	159	22
CuFe(CN) ₆	~5	/	20 mM ZnSO ₄	~1.73	53	92	23
CoFe(CN) ₆	/	/	4 M Zn(CF ₃ SO ₃) ₂	~1.75	173.4	303	24
I ₂	2.36	0.6	30 m ZnCl ₂	~1.48	612.5	905	25
LF-PLSD	4	/	1 M Zn(TFSI) ₂	~0.63	1148	725	26
Se	4~10	2.5	4 M Zn(CF ₃ SO ₃) ₂ + PEO	~1.1	664.7	729	27
Se	~1 and ~5.2	1	1 M ZnSO ₄	~1.23	611	751	28
TP-Se	~2.2	/	1 M Zn(OTf) ₂	~1.95	72.9	142	29
Te	1~1.5	1.1	30 m ZnCl ₂ + PEO	~0.84	1223.9	1028	30
Te	~1.5	/	1 M ZnSO ₄	~0.58	419	241	31

Te	~1.5	2	30 m ZnCl ₂	~0.83	802.7	~666	32
----	------	---	------------------------	-------	-------	------	----

To Reviewer 3:

This study presents an innovative six-electron Se conversion cathode chemistry for aqueous zinc batteries. The demonstrated Zn//Se cells show remarkable capacity and energy density, making a significant contribution to the field. Moreover, the introduction of a “dead-selenium revitalizer” strategy is particularly noteworthy, as it offers a solution to the challenge of capacity decay due to shuttling issues, which could have broader implications for other battery systems facing similar challenges. The manuscript is well-written, and the data comprehensively supports the conclusions. This work is suggested for publication in Nature Communications, pending the authors’ attention to the following issues.

Response: We appreciate the positive comment of the reviewer. Additional experiments and discussions have been conducted to address the following concerns.

1. The claim and comparison of energy density should be rigorous and objective. The authors are suggested to clearly state their calculation method for their energy density and confirm all values of listed energy density are calculated using the same standard. An estimated energy density based on the total mass of the cell is more convincing.

Response: Thank you for pointing out the inconsistency in our performance comparison. According to your suggestion, we made the necessary corrections to the relevant figures and table to ensure that all comparisons are based on the same standard. Specifically, the metrics shown in **Fig. R18a** (i.e., **Fig. 5b** in the manuscript) and **Table R3** (i.e., **Supplementary Table S1** in the manuscript) were all calculated based solely on the mass of active cathode materials. Additionally, the energy densities shown in **Fig. R18b** (i.e., **Fig. 5c** in the manuscript) were calculated based on the overall cell reaction, incorporating the masses of all active species, including the active anode material, cathode material, and any electrolyte salt involved in the electrochemical reaction. Additionally, the energy density of the Zn//Se cell based on the overall cell reaction is 404.2 Wh kg^{-1} . For practical device fabrication, significant system optimization efforts are required, particularly in terms of the N/P ratio, electrolyte volume, and the mass of inactive materials. According to a recent evaluation, the total mass of active materials in pouch cells should ideally constitute approximately 30% of the total cell mass³⁸. Based on this assumption, the energy density of the Zn//Se cell is projected to reach 121.3 Wh kg^{-1} when calculated using the total cell mass.

Fig. R18 a Comparison of performance (encompassing gravimetric capacity, average discharge voltage, and gravimetric energy density) of six-electron Se conversion chemistry with reported chalcogen cathodes for AZBs. The specific capacities and energy densities are calculated based on the mass of active cathode materials. **b** Comparison of energy density between the Zn//Se device and recently reported AZB devices. The presented energy densities were calculated based on the overall cell reaction, incorporating the masses of all active species, including the active anode material, cathode material, and any electrolyte salt involved in the electrochemical reaction.

Table R3 Performance comparison with the reported AZB cathode materials.

Cathode materials	Electrolyte	Midpoint voltage (V vs. Zn)	Specific capacity (mAh g ⁻¹)	Energy density (Wh kg ⁻¹)	Ref.
Se (22.7% in Se-AC compound)	30 m ZnCl ₂ + 0.1 m Et ₄ NBr + PEO	~1.03	2077	2138	This work
Se (75.5% in Se-AC compound)	30 m ZnCl ₂ + 0.1 m Et ₄ NBr + PEO	~1.14	1603	1823	This work
α -MnO ₂	2 M ZnSO ₄ + 0.1 M MnSO ₄	~1.31	285	373	9
β -MnO ₂	3 M Zn(CF ₃ SO ₃) ₂ + 0.1 M Mn(CF ₃ SO ₃) ₂	~1.13	225	254	10
K _{0.8} Mn ₈ O ₁₆	2 M ZnSO ₄ + 0.1 M MnSO ₄	~1.24	320	398	11
γ -MnO ₂	1 M ZnSO ₄	~1.25	285	356	12
V ₂ O ₅	3 M Zn(CF ₃ SO ₃) ₂	~0.58	470	273	13
V ₃ O ₇ •H ₂ O	1 M ZnSO ₄	~0.64	375	240	14
Zn _{0.3} V ₂ O ₅ •1.5H ₂ O	3 M Zn(CF ₃ SO ₃) ₂	~0.79	426	337	15
Calix[4]quinone	3 M Zn(CF ₃ SO ₃) ₂	~1	335	335	16
Para-dinitrobenzene	3 M Zn(CF ₃ SO ₃) ₂	~0.57	402	230	17
Polyaniline	1 M Zn(CF ₃ SO ₃) ₂	~1.1	200	220	18
3, 7- bis(phenylamino)phenothiazin-5- ium iodide	2.0 M ZnSO ₄	~1.1	188	207	19
1,4,5,8,9,12-hexaazatriphenylene- based COFs.	2.0 M ZnSO ₄	~0.84	344	289	20
Na ₃ V ₂ (PO ₄) ₃	0.5 M Zn(CH ₃ COO) ₂	~1.1	97	107	21
FeFe(CN) ₆	1 M Zn(OAc) ₂	~1.3	122	159	22
CuFe(CN) ₆	20 mM ZnSO ₄	~1.73	53	92	23
CoFe(CN) ₆	4 M Zn(CF ₃ SO ₃) ₂	~1.75	173.4	303	24
I ₂	30 m ZnCl ₂	~1.48	612.5	905	25

LF-PLSD	1 M Zn(TFSI) ₂	~-0.63	1148	725	²⁶
Se	4 M Zn(CF ₃ SO ₃) ₂ + PEO	~-1.1	664.7	729	²⁷
Se	1 M ZnSO ₄	~-1.23	611	751	²⁸
TP-Se	1 M Zn(OTF) ₂	~-1.95	72.9	142	²⁹
Te	30 m ZnCl ₂ + PEO	~-0.84	1223.9	1028	³⁰
Te	1 M ZnSO ₄	~-0.58	419	241	³¹
Te	30 m ZnCl ₂	~-0.83	802.7	~666	³²

2. Since the bromide redox couple plays a critical role as the inactive "dead" Se revitalizer to stabilize the electrode performance. I wonder if this is also applicable to aqueous sulfur batteries or if iodide redox serves a similar function. If this strategy applies to various multi-electron conversion-type cathodes, this work would be more impactful.

Response: We appreciate the insightful comment from the reviewer. In fact, the reported aqueous sulfur batteries predominantly rely on the S^0/S^{2-} conversion. Triggering the S^0/S^{4+} conversion in aqueous electrolytes remains a significant challenge due to the high potential required. As a result, the shuttling of polysulfide species (S_n^-) is more likely to lead to further reduction to S^{2-} on the anode surface, instead of forming dead S, which necessitates an oxidative reaction. Regarding the I-based redox couple, it indeed holds the potential to serve a similar function. However, its mechanism would differ from the Br-based couple. The I^-/I_n^- couple (0.535 V vs. SHE) exhibits a lower potential compared to the Se^0/Se^{4+} conversion (0.739 V vs. SHE). On the other hand, the I^0/I^+ couple could prove helpful due to its suitable potential (1.22 V vs. SHE). We also agree with the reviewer that the reported concept of the dead-material revitalizer could be further extended to different multi-electron conversion battery systems. We are currently conducting several related studies to explore this potential.

3. In the XPS data of the pristine Se^0 state, there are distinct peaks labeled Se-C that show higher binding energy compared to Se^0 . The authors should provide a more detailed explanation for why these peaks appear in the pristine sample.

Response: The slightly higher binding energy of the Se-C peaks in the pristine Se electrode can be attributed to the fabrication process of the Se-activated carbon (AC) compound. Se-AC was prepared through melt-diffusion process of Se into AC, which triggered the formation of interfacial Se-C bonds and induced electron transfer from Se to C. A similar phenomenon was previously observed in the Se-carbon composite prepared by melt-diffusion method³⁹. During the Se conversion process, these interfacial Se-C bonds disappeared, thus leading to dominant Se^0 peak. The corresponding discussion has been added to the revised manuscript (**Page 7, Paragraph 3**).

4. The XPS data suggests a possible phase change after Se melting in AC. To provide a more comprehensive characterization, please include XRD patterns of both the pristine Se and the Se-AC composite.

Response: Thank you for the valuable suggestion. The XRD patterns of pristine Se, Se-AC composites

with both low Se content (22.7%, denoted Se-AC-L) and high Se content (75.5%, denoted Se-AC-H) have been collected and are shown in **Fig. R19**. Both Se-AC-L and Se-AC-H show no characteristic peaks associated with Se, suggesting the amorphous nature of Se in both samples. The corresponding discussion has been added to the revised manuscript (**Page 5, Paragraph 2; Page 16, Paragraph 2**).

Fig. R19 XRD spectra of pristine Se, Se-AC-L, and Se-AC-H.

5. The preparation process of the quasi-solid electrolyte is not clearly explained, which is crucial for readers to replicate your experiments. The authors are suggested to include some optical images that illustrate the steps involved in preparing the quasi-solid electrolyte.

Response: To address your concern, we took optical images that illustrate the steps involved in preparing the quasi-solid electrolyte (QSE) (**Fig. R20**). The glass fiber separator (Whatman GF/C, 10 mm in diameter; 0.2 mm in thickness) (**Fig. R20a**) was first coated with graphene oxide (GO) with around 0.25 mg cm^{-2} through vacuum filtration (**Fig. R20b**). Next, the separator was immersed in a hydrogel composed of 30 m ZnCl_2 , 0.1 m Et_4NBr , and 10 wt% PEO at 80 °C overnight. Afterward, the excess hydrogel on the separator surface was removed, and the separator was dried in an oven at 80 °C for 6 h (**Fig. R20c**). For electrochemical measurements with ZCE, the separator was treated in a similar way, except using a hydrogel composed of 30 m ZnCl_2 and 10 wt% PEO. The corresponding discussion has been added to the revised manuscript (**Page 18, Paragraph 2**).

Fig. R20 Digital photos of **a** commercial Whatman glass fiber membrane, **b** GO decorated membrane and **c** QSE plate.

6. Why was YP80 AC chosen as the host material? Did the author explore other host materials for Se? What impact do different host materials have on improving the stability of the cells? These issues should be further clarified in the manuscript.

Response: We selected commercial YP80 AC as the standard host material for Se due to its high specific surface area ($2271 \text{ m}^2 \text{ g}^{-1}$) and large porosity ($>0.8 \text{ mL g}^{-1}$), which provide ample space to accommodate active Se species. Additionally, the conductive nature of AC facilitates efficient charge transport, ensuring high utilization efficiency of the active material. We agree with the reviewer that optimizing host materials is crucial for further enhancing the performance of Zn//Se batteries, particularly by developing advanced porous structures with strong confinement capabilities. While this is an important direction for future research, it falls outside the scope of this study. The corresponding discussion has been added to the revised manuscript (**Page 17, Paragraph 3**).

7. In Fig. 5a, the discharge capacity of the Zn//Se cell is indicated as 2077 mAh/g. The authors estimate that after subtracting the capacity contribution of AC, a capacity of 1907.6 mAh/g is attributed to the Se conversion reaction. However, there is likely some contribution from the Br^0/Br^- conversion as well. The authors should account for this contribution to determine the true capacity of the Se conversion reaction more accurately.

Response: We are grateful to the reviewer for pointing out this issue. Indeed, the $\text{Br}^-/\text{Br}_n^-$ couple could also contribute to the capacity of the Zn//Se cell with ZCE-Br. To exclude this capacity contribution, we assembled a Zn//AC cell with ZCE-Br and derived the specific capacity of the AC electrode (101.8 mAh g^{-1}) from the GCD measurement (**Fig. R21**). This specific capacity includes the contribution from the $\text{Br}^-/\text{Br}_n^-$ couple and the capacitive charge storage of AC. Based on this result, the specific capacity associated with Se conversion was estimated to be $1731.5 \text{ mAh g}^{-1}$. The corresponding discussion has been added to the revised manuscript (**Page 15, Paragraph 2**).

Fig. R21 GCD curves of the Zn//Se cell with ZCE-Br.

References:

1. Robinson E, Ciruna J. The chlorosulfuric acid solvent system. Part II. The solutes SeCl_4 and TeCl_4 ; evidence for the formation of the SeCl_3^+ and TeCl_3^+ ions. *Can. J. Chem.* **46**, 3197-3200 (1968).
2. Murchie MP, Passmore J, White PS. The characterisation and X-ray crystal structure of pentabromodiselenium hexafluoroarsenate, $\text{Se}_2\text{Br}_5\text{AsF}_6$; some thermodynamic considerations and the nonexistence of $\text{Se}_2\text{I}_5\text{AsF}_6$. *Can. J. Chem.* **65**, 1584-1593 (1987).
3. Ji X. A perspective of ZnCl_2 electrolytes: the physical and electrochemical properties. *eScience* **1**, 99-107 (2021).
4. Torrie B. Raman and Infrared Spectra of Na_2SeO_3 , NaHSeO_3 , H_2SeO_3 , and $\text{NaH}_3(\text{SeO}_3)_2$. *Can. J. Phys.* **51**, 610-615 (1973).
5. Zhang Q, Li H, Ma Y, Zhai T. ZnSe nanostructures: synthesis, properties and applications. *Prog. Mater. Sci.* **83**, 472-535 (2016).
6. Mahmood A, Zheng Z, Chen Y. Zinc–Bromine Batteries: Challenges, Prospective Solutions, and Future. *Adv. Sci.* **11**, 2305561 (2024).
7. Lin S, *et al.* Zn Anode Surviving Extremely Corrosive Polybromide Environment with Alginate-Graphene Oxide Hydrogel Coating. *Small* **20**, 2311510 (2024).
8. Chen C, Zhang J, Hu B, Liang Q, Xiong X. Dynamic gel as artificial interphase layer for ultrahigh-rate and large-capacity lithium metal anode. *Nat. Commun.* **14**, 4018 (2023).
9. Pan H, *et al.* Reversible aqueous zinc/manganese oxide energy storage from conversion reactions. *Nat. Energy* **1**, 16039 (2016).
10. Zhang N, *et al.* Rechargeable aqueous zinc-manganese dioxide batteries with high energy and power densities. *Nat. Commun.* **8**, 405 (2017).
11. Fang G, *et al.* Suppressing manganese dissolution in potassium manganate with rich oxygen defects engaged high-energy-density and durable aqueous zinc-ion battery. *Adv. Funct. Mater.* **29**, 1808375 (2019).
12. Alfuruqi MH, *et al.* Electrochemically induced structural transformation in a $\gamma\text{-MnO}_2$ cathode of a high capacity zinc-ion battery system. *Chem. Mater.* **27**, 3609-3620 (2015).
13. Zhang N, *et al.* Rechargeable aqueous Zn– V_2O_5 battery with high energy density and long cycle life. *ACS Energy Lett.* **3**, 1366-1372 (2018).

14. Kundu D, Hosseini Vajargah S, Wan L, Adams B, Prendergast D, Nazar LF. Aqueous vs. nonaqueous Zn-ion batteries: consequences of the desolvation penalty at the interface. *Energy Environ. Sci.* **11**, 881-892 (2018).
15. Wang LL, Huang KW, Chen JT, Zheng JR. Ultralong cycle stability of aqueous zinc-ion batteries with zinc vanadium oxide cathodes. *Sci. Adv.* **5**, 10 (2019).
16. Zhao Q, *et al.* High-capacity aqueous zinc batteries using sustainable quinone electrodes. *Sci. Adv.* **4**, 10 (2018).
17. Song Z, *et al.* Anionic co-insertion charge storage in dinitrobenzene cathodes for high-performance aqueous zinc-organic batteries. *Angew. Chem. Int. Ed.* **61**, e202208821 (2022).
18. Wan F, Zhang L, Wang X, Bi S, Niu Z, Chen J. An aqueous rechargeable zinc-organic battery with hybrid mechanism. *Adv. Funct. Mater.* **28**, 1804975 (2018).
19. Wang N, *et al.* Molecular tailoring of an n/p-type phenothiazine organic scaffold for zinc batteries. *Angew. Chem. Int. Ed.* **60**, 20826-20832 (2021).
20. Wang W, *et al.* Molecular engineering of covalent organic framework cathodes for enhanced zinc-ion batteries. *Adv. Mater.* **33**, 2103617 (2021).
21. Li G, *et al.* Towards polyvalent ion batteries: A zinc-ion battery based on NASICON structured $\text{Na}_3\text{V}_2(\text{PO}_4)_3$. *Nano Energy* **25**, 211-217 (2016).
22. Liu Z, Pulletikurthi G, Endres F. A Prussian blue/zinc secondary battery with a bio-Ionic liquid-water mixture as electrolyte. *ACS Appl. Mater. Interfaces* **8**, 12158-12164 (2016).
23. Trocoli R, La Mantia F. An aqueous zinc-ion battery based on copper hexacyanoferrate. *ChemSusChem* **8**, 481-485 (2015).
24. Ma L, *et al.* Achieving high-voltage and high-capacity aqueous rechargeable zinc Ion battery by incorporating two-species redox reaction. *Adv. Energy Mater.* **9**, 1902446 (2019).
25. Liang G, *et al.* Development of rechargeable high-energy hybrid zinc-iodine aqueous batteries exploiting reversible chlorine-based redox reaction. *Nat. Commun.* **14**, 1856 (2023).
26. Zhao Y, *et al.* Initiating a reversible aqueous Zn/sulfur battery through a "Liquid Film". *Adv. Mater.* **32**, e2003070 (2020).
27. Ma L, *et al.* Electrocatalytic Selenium Redox Reaction for High-Mass-Loading Zinc-Selenium Batteries with Improved Kinetics and Selenium Utilization. *Adv. Energy Mater.* **12**, 2201322 (2022).

28. Chen Z, *et al.* Zinc/selenium conversion battery: a system highly compatible with both organic and aqueous electrolytes. *Energy Environ. Sci.* **14**, 2441-2450 (2021).
29. Chen Z, *et al.* Anion chemistry enabled positive valence conversion to achieve a record high-voltage organic cathode for zinc batteries. *Chem* **8**, 2204-2216 (2022).
30. Du J, *et al.* A High-Energy Tellurium Redox-Amphoteric Conversion Cathode Chemistry for Aqueous Zinc Batteries. *Adv. Mater.*, 2313621 (2024).
31. Chen Z, *et al.* Aqueous zinc-tellurium batteries with ultraflat discharge plateau and high volumetric capacity. *Adv. Mater.* **32**, e2001469 (2020).
32. Chen Z, *et al.* Tellurium with Reversible Six-Electron Transfer Chemistry for High-Performance Zinc Batteries. *J. Am. Chem. Soc.* **145**, 20521-20529 (2023).
33. Dai C, *et al.* Fast constructing polarity-switchable zinc-bromine microbatteries with high areal energy density. *Sci. Adv.* **8**, eabo6688 (2022).
34. Wang C, *et al.* Visualizing and Understanding the Ionic Liquid-Mediated Polybromide Electrochemistry for Aqueous Zinc-Bromine Redox Batteries. *Nano Lett.* **24**, 13796-13804 (2024).
35. Qiu J, *et al.* Enabling stable cycling of 4.2 V high-voltage all-solid-state batteries with PEO-based solid electrolyte. *Adv. Funct. Mater.* **30**, 1909392 (2020).
36. Tian G, *et al.* Study of the Lithium Storage Mechanism of N-Doped Carbon-Modified Cu₂S Electrodes for Lithium-Ion Batteries. *Chem. Eur. J.* **27**, 13774-13782 (2021).
37. Kim H, Yeo S, Kim M, Lee G. Advancing silicon-based Li-ion batteries: enhanced stability and performance through carbon-coated Si and rGO linkage. *J. Mater. Sci.* **58**, 13621-13634 (2023).
38. Liu S, *et al.* Zinc ion Batteries: Bridging the Gap from Academia to Industry for Grid-Scale Energy Storage. *Angew. Chem. Int. Ed.* **63**, e202400045 (2024).
39. Kim JK, Kang YC. Encapsulation of Se into hierarchically porous carbon microspheres with optimized pore structure for advanced Na–Se and K–Se batteries. *ACS nano* **14**, 13203-13216 (2020).

To Reviewer 1:

My previous comments have been addressed by the authors. I recommend publication of this revised manuscript.

Response: We sincerely appreciate your positive feedback and recommendation for publication.

To Reviewer 2:

After carefully reconsidering the manuscript based on the authors' responses to my review comments, I find that, while the authors have made efforts to address the concerns raised, the manuscript does not fully meet the rigorous standards expected for publication in a decent journal like Nature Communications.

Response: We appreciate the valuable comment of the reviewer. However, we politely disagree with the comments regarding the novelty and significance of the work. We have provided detailed arguments in the point-by-point responses. Moreover, we would like to emphasize that both Reviewer 1 and Reviewer 3 acknowledge our study for introducing a 'noteworthy dead-selenium revitalizer strategy' and 'making a significant contribution to the field.'

Firstly, recent literature has reported that superhalide-anion-containing electrolytes can activate the six-electron conversion of selenium (10.1021/acs.nanolett.4c00198), indicating that the $\text{Se}^{2-}/\text{Se}^0/\text{Se}^{4+}$ process is not novel. Although the Zn//Se battery achieved higher capacities compared to previous systems, this improvement appears to be incremental rather than a fundamental breakthrough.

Response: In our work, we demonstrate the reversible six-electron $\text{Se}^{2-}/\text{Se}^0/\text{Se}^{4+}$ via a novel ZnSe/Se/SeCl₄ process, marking the first time that this specific process can be achieved for aqueous batteries. While we appreciate the reviewer for bringing the referenced literature to our attention, we believe it is not fair to judge the novelty of our work by comparing it with a Se conversion chemistry triggered in a fundamentally different electrolyte system, namely non-aqueous ionic liquids (EMImCl/ZnCl₂/ZnF₂ in the referenced work). In the ionic liquid electrolyte, the Se electrode could only achieve a specific capacity of $\sim 700 \text{ mAh g}^{-1}$, reflecting a low Se utilization efficiency of only 34%. In stark contrast, our Se electrode significantly

outperforms this benchmark by delivering a high specific capacity of 1767.3 mAh g⁻¹. Additionally, our study uniquely introduces the innovative concept of a dead-material revitalizer, which further distinguishes our work and underscores its novelty and impact. We have added the corresponding discussion in the revised manuscript (**Page 8, Paragraph 1**).

Secondly, the concept of a dead-material revitalizer, mentioned by the authors, is not new and has been reported in other types of batteries (10.1038/s41560-021-00789-7; 10.1002/anie.202110589).

Response: Our study introduces dead-material revitalizers as a promising and novel approach for stabilizing conversion-type cathodes. Specifically, we disclose that the incorporation of the Br⁻/Br_n⁻ redox couple into the Zn//Se cell enables the generated Br_n⁻ species to act as a dead-Se revitalizer. These species react with the Se passivation layer on the Zn anode, regenerating active Se for cathode reactions. The two references cited by the reviewer focus on Li metal anodes, where inactive Li metal results from the electrical isolation of Li dendrites. In those cases, inactive Li metal was reclaimed by introducing I₃⁻ to chemically react with the inactive Li. Clearly, these reported cases pertain to entirely different battery systems and involve fundamentally different reaction mechanisms.

Thirdly, the long-term cycling stability of the Zn//Se battery in this work falls short of previously reported aqueous Zn//Se batteries (10.1039/D0EE02999H; 10.1002/advs.202403224; 10.1002/aenm.202201322). Furthermore, the failure mechanism of the system and the long-term stability of the Zn//Se battery require more extensive exploration.

Response: We appreciate the comment of the reviewer, but we believe that it is not fair to evaluate our work solely based on cycling stability while disregarding the completely different conversion mechanism and significant performance advantages of our Zn//Se cell compared to the referenced studies. Specifically, the Se electrode in the first reference (10.1039/D0EE02999H) relies on a Se/ZnSe conversion, achieving only a limited specific capacity of 611 mAh g⁻¹ and exhibiting a large voltage polarization of 490 mV. The Se electrodes in both the second (10.1002/advs.202403224) and the third

(10.1002/aenm.202201322) references operate through a ZnSe/Se/SeO₂ conversion mechanism. Due to the sluggish conversion kinetics in these cases, both Se electrodes show limited Se utilization efficiencies (29.6% and 32.6%) and large voltage polarization (1070 mV and 540 mV). In contrast, our study introduces a novel Se⁰/Se⁴⁺ conversion mechanism leveraging the relatively small, single-atom Cl⁻ anions as charge carriers. This mechanism enables our Se electrode to achieve a high Se utilization efficiency (up to 86.7%) and a significantly reduced charge/discharge polarization (140 mV). These metrics highlight the unique and substantial advances in our work, which are distinct from the referenced studies.

While we acknowledge that the cycling stability of the Se cathode in our system still has room for improvement, we have already added further discussion in our last-round revision to analyze possible solutions. These include the development of advanced porous hosts with strong confinement effects and the use of coated interphases or functionalized separators to inhibit the dissolution of active Se species. However, we also emphasize that fully addressing these challenges requires extensive future research on system optimization, rather than expecting a single study to solve all issues. Significant inspiration can be drawn from research on non-aqueous rechargeable Li-S batteries, which began in the 1960s and continues to be a highly active field. Similarly, we believe the findings in this study provide a valuable foundation and critical insights to guide future advancements in aqueous Zn//Se batteries.

Given these considerations, while the manuscript contains valuable insights, I believe it would be more suitable for submission to a more specialized journal where these issues can be explored in greater depth. Additionally, there are a few minor issues in the paper that require clarification:

Response: We thank the reviewer for taking the time to provide the valuable comment. We have carefully revised the manuscript by including additional experiments and expanded discussions. As reflected in the shared views of the other two reviewers, we firmly believe that the revised manuscript meets the high standards of *Nature Communications*.

1. The Se-C bond is present in the pristine sample in Figure 3c, but it disappears during the

charge/discharge process. Is the Se-C bond converted during this process, and if so, how does this affect the electrochemical performance?

Response: We appreciate the insightful question. The Se-C bond observed in the pristine Se electrode arises from the high-temperature melt-diffusion process (at 600 °C) of Se into activated carbon during electrode preparation. During the initial charging process, Se is oxidized to SeCl₄, resulting in the disappearance of the Se-C bond. In subsequent cycles, the conditions required for the reformation of the Se-C bond, such as the high-temperature melt-diffusion process, are absent. Therefore, the disappearance of the Se-C bond has no impact on the electrochemical performance of the system. We have added the corresponding discussion to the revised manuscript (**Page 7, Paragraph 2**).

2. Both Supplementary Figure 4a and Supplementary Figure 16b should depict the charge/discharge curves of Zn//AC batteries, but the curves in these two figures differ, with a discharge platform clearly visible at 1.4 V in Supplementary Figure 16b.

Fig. R1 The initial three-cycle GCD profiles of the Zn//AC cell with ZCE at 0.2 A g⁻¹.

Response: Thank you for the valuable comment. To address this concern, we have repeated the measurement in **Supplementary Figure 16b**. As presented in **Fig. R1**, the obtained curves are almost identical to our previous results, exhibiting quasi-slope characteristics. The slight platform-like shape observed could be attributed to redox-like behavior associated with ion solvation/desolvation process¹⁻³ or surface functional groups on activated carbon⁴⁻⁶. The shape deviation between **Supplementary Figure 16b** and **Supplementary Figure 4a** can be

explained by the difference in voltage windows used in the two tests. We have added the corresponding explanation to the revised supplementary materials.

To Reviewer 3:

The authors have done considerable work to address the reviewers' concerns. The revised manuscript is acceptable for publication in Nature Communications. I have only minor further revision comments to improve the academic rigor of this paper. The authors are suggested to report the potential of the positive electrodes, not voltage, as appropriate electrochemistry terminology, as Nat. Nanotechnol. called upon recently (<https://www.nature.com/articles/s41565-024-01844-6>). Additionally, the reference redox couple should be “Zn²⁺/Zn”, not just “Zn”.

Response: We appreciate the valuable comment and the recommendation for publication. Following your suggestion, we have updated the expression of the device voltage (vs. Zn) to the electrode potential (vs. Zn²⁺/Zn). Detailed changes can be found in the revised manuscript.

References:

1. Ge K, Shao H, Raymundo-Piñero E, Taberna P-L, Simon P. Cation desolvation-induced capacitance enhancement in reduced graphene oxide (rGO). *Nat. Commun.* **15**, 1935 (2024).
2. Hu L, Guo D, Feng G, Li H, Zhai T. Asymmetric behavior of positive and negative electrodes in carbon/carbon supercapacitors and its underlying mechanism. *J. Phys. Chem. C* **120**, 24675-24681 (2016).
3. Urita K, Ide N, Isobe K, Furukawa H, Moriguchi I. Enhanced electric double-layer capacitance by desolvation of lithium ions in confined nanospaces of microporous carbon. *ACS nano* **8**, 3614-3619 (2014).
4. Hu YR, Dong XL, Zhuang HK, Yan D, Hou L, Li WC. Introducing Electrochemically Active Oxygen Species to Boost the Pseudocapacitance of Carbon-based Supercapacitor. *ChemElectroChem* **8**, 3073-3079 (2021).
5. Qiu C, Jiang L, Gao Y, Sheng L. Effects of oxygen-containing functional groups on carbon materials in supercapacitors: A review. *Mater. Design* **230**, 111952 (2023).
6. Liu F, Xue D. An electrochemical route to quantitative oxidation of graphene frameworks with controllable C/O ratios and added pseudocapacitances. *Chem. Eur. J.* **19**, 10716-10722 (2013).